# VADUGS: A neural network for the remote sensing of volcanic ash with MSG/SEVIRI trained with synthetic thermal satellite observations simulated with a radiative transfer model

Luca Bugliaro[1], Dennis Piontek[1], Stephan Kox[1,2], Marius Schmidl[1,3], Bernhard Mayer[4,1], Richard Müller[5], Margarita Vázquez-Navarro[1,6], Daniel M. Peters[7,8], Roy G. Grainger[9], Josef Gasteiger[4,10], and Jayanta Kar[11,12]

[1]Deutsches Zentrum für Luft- und Raumfahrt (DLR), Institut für Physik der Atmosphäre, Oberpfaffenhofen, Germany
[2]now at: Telespazio Germany GmbH, Darmstadt, Germany
[3]now at: MTU Aero Engines AG
[4]Ludwig-Maximilians Universität, Meteorologisches Institut, München, Germany
[5]Deutscher Wetterdienst, Offenbach, Germany
[6]now at: European Organisation for the Exploitation of Meteorological Satellites (EUMETSAT), Darmstadt, Germany
[7]Atmospheric, Oceanic and Planetary Physics, University of Oxford, Oxford, UK
[8]now at: RAL Space, STFC Rutherford Appleton Laboratory, Harwell, UK
[9]COMET, Atmospheric, Oceanic and Planetary Physics, University of Oxford, Oxford, UK
[10]now at: University of Vienna, Faculty of Physics, Vienna, Austria
[11]Science System and Applications, Inc., Hampton, VA, USA
[12]Science Directorate, NASA Langley Research Center, Hampton, VA, USA

**Correspondence:** Luca Bugliaro (luca.bugliaro@dlr.de)

**Abstract.** After the eruption of volcanoes all over the world the monitoring of the dispersion of ash in the atmosphere is an important task for satellite remote sensing since ash represents a threat to air traffic. In this work we present a novel method, tailored for Eyjafjallajökull ash but applicable to other eruptions as well, that uses thermal observations of the SEVIRI imager aboard the geostationary Meteosat Second Generation satellite to detect ash clouds and determine their mass column

concentration and top height during day and night. This approach requires the compilation of an extensive data set of synthetic SEVIRI observations to train an artificial neural network. This is done by means of the RTSIM tool that combines atmospheric, surface and ash properties and runs automatically a large number of radiative transfer calculations for the entire SEVIRI disk. The resulting algorithm is called VADUGS (Volcanic Ash Detection Using Geostationary Satellites) and has been evaluated against independent radiative transfer simulations. VADUGS detects ash contaminated pixels with a probability of detection

of 0.84 and a false alarm rate of 0.05. Ash column concentrations are provided by VADUGS with correlations up to 0.5, a scatter up to $0.6\,\mathrm{g\,m^{-2}}$ for concentrations smaller than $2.0\,\mathrm{g\,m^{-2}}$ and small overestimations in the range 5–50 % for moderate viewing angles 35–65°, but up to 300 % for satellite viewing zenith angles close to 90° or 0°. Ash top heights are mainly underestimated, with the smallest underestimation of -9 % for viewing zenith angles between 40° and 50°. Absolute errors are smaller than 70 % and with high correlation coefficients up to 0.7 for ash clouds with high mass column concentrations. A

comparison against spaceborne lidar observations by CALIPSO/CALIOP confirms these results: For six overpasses over the ash cloud from the Puyehue-Cordón Caulle volcano in June 2011, VADUGS shows similar features as the corresponding lidar

data, with a correlation coefficient of 0.49 and an overestimation of ash column concentration by 55 %, although still in the range of uncertainty of CALIOP. A comparison to another ash algorithm shows that both retrievals provide plausible detection results, with VADUGS being able to detect ash further away from the Eyjafjallajökull volcano, but sometimes missing the thick ash clouds close to the vent. VADUGS is run operationally at the German Weather Service and this application is presented as well.

*Copyright statement.* TEXT

# 1 Introduction

Volcanic ash is a threat for air traffic, as it can damage aircraft engines and lead to flame-outs, see e.g. Miller and Casadevall (2000). The eruption of the volcano Eyjafjallajökull in 2010 had a huge impact on the European air traffic causing a massive cancellation of flights (see e.g. Budd et al., 2011; Langmann et al., 2012). As a consequence, the interest in tracking volcanic ash clouds in case of a similar scenario increased. Satellite images are particularly useful in this context because of their large field of regard and often high temporal or spatial resolution. In particular, observations of the SEVIRI radiometer aboard Meteosat Second Generation (MSG) are very well suited due to its 12 spectral channels (four solar, seven thermal and a mixed channel) that scan the Earth from above the Equator every 15 minutes with a spatial sampling distance of 3 km at nadir. Various methods to detect volcanic ash plumes in satellite images have been developed in the last years (Prata, 1989b; Francis et al., 2012; Mackie and Watson, 2014; Piscini et al., 2014; Guéhenneux et al., 2015; Gouhier et al., 2020, among others). Many of them exploit thermal observations since they provide both daytime and nighttime coverage and because they show a particular signature of many volcanic ash types, the "reverse" absorption feature (Wen and Rose, 1994; Pavolonis et al., 2006). It states that the brightness temperature difference $BTD(\lambda_1 - \lambda_2)$ between the SEVIRI channel centred at $\lambda_1 = 10.8\,\mu$m and the SEVIRI channel centred at $\lambda_2 = 12.0\,\mu$m has the opposite sign as the same BTD for ice clouds, thus enabling the identification of volcanic ash contaminated pixels.

Our approach focuses on the MSG/SEVIRI sensor and aims at gaining information about volcanic ash clouds through the application of an artificial neural network (NN). Before a NN using the supervised learning approach can be used to solve this remote sensing problem, training data has to be provided for its learning phase during which the NN learns to approximate the desired volcanic ash properties from the given input data / satellite observations. For volcanic eruptions, there are no documented situations in which the actual three dimensional (in space) or four dimensional (in space and time) mass distribution of the volcanic aerosol is known. In-situ measurements are rare, can probe only a limited part of the ash clouds at selected locations and at particular times, and are difficult to compare with passive satellite observations. Training data have been composed by combining satellite images with models/simulations/retrievals, e.g. Gray and Bennartz (2015, the polar orbiting MODIS reflectometer and HYSPLIT trajectories), Picchiani et al. (2011, MODIS and look-up tables), Piscini et al. (2014, MODIS and other retrievals), Zhu et al. (2020, SEVIRI and CALIOP lidar observations). As an alternative, simulated

satellite data can be used. These consist of simulated brightness temperatures (BTs) a satellite instrument viewing the Earth's atmosphere would measure in given situations with and without generic volcanic ash layers. Using these synthetic obsevations,

a suitably designed NN can be applied to generalize the relationship between input (simulated satellite observations) and output (volcanic ash properties). After training, the NN should be able to derive the latter for a given set of measured BTs. This principle has been utilised for a retrieval of the mass column density and the cloud top height of volcanic ash. The resulting algorithm, called VADUGS (Volcanic Ash Detection Using Geostationary Satellites), is presented in this manuscript. It has been developed in the aftermath of the impressive eruption of the Icelandic Eyjafjallajökull volcano in 2010 according to the

experience gathered with ice clouds in Kox et al. (2014) and with particular focus on this eruption. It was developed for the investigation of the Eyjafjallajökull eruption and for the detection of future Eyjafjallajökull-like eruptions that might have similar impacts on airtraffic over Europe as the 2010 event. It was first presented at the 2013 EUMETSAT Conference held in Vienna, Austria (Kox et al., 2013). Since 2015 VADUGS is run operationally at the German Weather Service (Deutscher Wetterdienst, DWD, DWD2015) and from 2016 to 2019 it provided satellite observations of volcanic ash for the EU Horizon

2020 project EUNADICS-AV (European Natural Airborne Disaster Information and Coordination System for Aviation, Brenot et al., 2021). VADUGS participated in two WMO intercomparison workshops for satellite products about volcanic ash in 2015 (Graf et al., 2015; WMO2015) and 2018 and its approach has been extended to Himawari-8 data (de Laat et al., 2020). VADUGS is also the basis for more advanced machine learning algorithms for detection of volcanic ash and the determination of its properties (Piontek et al., 2021c,b).

This article describes a) a method to generate comprehensive realistic data sets of simulated thermal observations considering both liquid and ice water clouds as well as volcanic ash and b) the development and performance of a NN trained using these synthetic data sets. In Sect. 2 the main concepts and ideas are introduced. The tool for the compilation of the simulated satellite observations is described in Section 3 and the resulting data set of synthetic observations is presented in Sect. 4. Sections 4.2 and 4.3 explain the training of the NN retrieval and present some considerations about its application together with an illustration

of the retrieval results for the Eyjafjallajökull May eruption phase. In Sect. 4.4 the validation against independent simulated observations and against spaceborne lidar measurements are shown, while the implementation at DWD is sketched in Sect. 5 before conclusions are drawn in Sect. 6.

## 2 Approach

The VADUGS retrieval (Volcanic Ash Detection Using Geostationary Satellites) has been developed closely following the

75 experience gathered with ice clouds in Kox et al. (2014). There, ice clouds are identified in MSG/SEVIRI thermal observations by means of a NN that has been trained with collocated cloud products from the CALIOP lidar in space (Winker et al., 2009). However, as already explained in the introduction, this approach cannot be directly applied to volcanic ash because volcanic ash observations in the field of regard of MSG/SEVIRI are limited to few large eruptions (Eyjafjallajökull in 2010, Puyehue-Cordón Caulle in 2011) and some smaller eruptions (e.g. Grimsvötn 2011, Etna, Nabro 2011) for which no comprehensive data

set is available containing both satellite observations and the corresponding ash properties that can be used to train a NN. Thus,

the approach selected for volcanic ash makes use of the radiative transfer model libRadtran (Emde et al., 2016; Mayer and Kylling, 2005) to compute one dimensional (1D) simulated satellite observations based on a data set of consistent but variable atmospheric properties, including ash layer height, geometrical thickness and mass concentration at different places inside the MSG/SEVIRI field of regard under realistic atmospheric conditions. For the derivation of top-of-atmosphere fluxes from MSG/SEVIRI observations a similar approach (Vázquez-Navarro et al., 2013) has already proved to provide good results. Thus, in this study we combine the NN technique from Kox et al. (2014) with the radiative transfer approach from Vázquez-Navarro et al. (2013).

The goal of the VADUGS retrieval algorithm is the derivation of volcanic ash cover (VAC), volcanic ash top height (VATH) and volcanic ash mass column concentration (VAMC).

The main advantages of using simulated satellite observations for the training of a volcanic ash retrieval is that 1) all atmospheric properties, including of course volcanic and liquid/ice cloud properties are exactly known, 2) the data set can cover the entire SEVIRI disk and all possible combinations of meteorological clouds, volcanic ash, time of the year / of the day and viewing geometries, 3) in principle, various ash types might be included. The main drawback is that radiative calculations, although realistic, might still not encompass the full complexity of real observations.

## 2.1 MSG/SEVIRI

Meteosat Second Generation (MSG) is a series of four satellites operated by EUMETSAT that has become operational in 2004 and lasts until now. The satellites, MSG1 to MSG4, have been renamed after launch to Meteosat-8 to Meteosat-11. Their main instrument is the Spinning Enhanced Visible and Infrared Imager (SEVIRI). This radiometer has 11 spectral channels from the visible to the infrared with a spatial sampling of 3 km at the subsatellite point and a broadband High Resolution Visible (HRV) channel with a spatial sampling distance of 1 km at nadir. The thermal channels are centred at 6.2 $\mu$m and 7.3 $\mu$m (strong water vapour absorption), 8.7 $\mu$m, 10.8 $\mu$m and 12.0 $\mu$m (window channels), as well as 9.7 $\mu$m (ozone absorption) and 13.4 $\mu$m (carbon dioxide absorption). The operational service provides full disk Earth data every 15 min and the rapid scan service observes the upper part of the Earth disk with Europe and North Africa with a repetition time of 5 min. For the development of VADUGS we concentrate on MSG2 (Meteosat-9), launched in 2005, since it was the prime Meteosat satellite from 2006 until 2013 and thus recorded the main volcanic eruptions mentioned above (Eyjafjallajökull, Grimsvötn, Puyehue).

## 2.2 VADUGS: an artificial neural network

An artificial NN (Rumelhart et al., 1986) consists of a set of neurons that exchange information with each other with the goal to derive a set of output variables given a set of known input quantities. The technical implementation of the NN in this study follows very closely Kox et al. (2014) since the NN developed there proved to be very effective in detecting ice clouds and in the determination of their properties.

The neurons in the NN are structured in three layers: (a) the input layer, (b) one hidden layer, and (c) the output layer. Correspondingly, neurons in these layers are called input neurons, hidden neurons and output neurons. Input neurons transport the information used for the detection of ash and the determination of its properties into the NN, i.e. each observation and

ancillary data is assigned a single neuron. Output neurons contain the information about the desired ash properties, while the hidden neurons collect, combine and process data forwarded by the input neurons and fire the results to the output neurons, where the output quantities are produced. The number of hidden neurons is selected in analogy to the ice cloud retrieval COCS (Kox et al., 2014) that proved to yield accurate results for the remote sensing of ice clouds with the same spaceborne sensor MSG2/SEVIRI and with 10 input and two output variables. As in Kox et al. (2014), 600 neurons for the hidden layer have been adopted as a trade-off between accuracy and CPU time consumption. Similarly, VADUGS uses 17 input variables (see Sect. 4.2) and two output variables, VAMC and VATH. A feed-forward NN is implemented where all connections between neurons are in the forward direction (from input layer to output layer through the hidden layer), while connections within a layer or backward connections are forbidden. A numeric tunable weight is assigned to each neuron connection. Every neuron processes the output from all neurons in the preceding layer weighted with the corresponding connection weights through a non-linear activation function, the logistic function in our case. Thus, the NN can learn patterns and approximate functions in a sort of multi-dimensional non-linear fitting by means of a limited number of neurons. Training the NN determines the value of all the connections weights through the backpropagation technique. The input variables are fed into the NN, which computes a vector of output variables using its current weights. The total squared difference error between the estimated output and the corresponding vector of expected output is propagated backwards through the NN to update each weight using gradient descent in order to minimise the total error. The training stops when no reduction of the sum of the quadratic deviations is observed.

With this technique an exhaustive set of training examples containing both input and output variables must be available.

## 3  Simulated satellite observations

In this section the details of the radiative transfer calculations are explained.

### 3.1  Radiative transfer calculations

Accurate and realistic one dimensional (1D) simulations of the satellite observations with realistic and representative atmospheric conditions are crucial for the successful training of the NN and especially for its successful application to real data.

#### 3.1.1  Scene selection

The MSG/SEVIRI grid consists of more than 13 million pixels, including space. In order to homogeneously cover the full Earth disk with a moderate amount of simulations and in view of the coarser spatial resolution of the atmospheric data from the European Centre for Medium-Range Weather Forecasts (ECMWF, see Sect. 3.1.2), the original satellite grid was reduced by a factor of 100 by selecting every tenth pixel in x and y direction. This results in a grid with 102,799 points that are homogeneously distributed over the SEVIRI disk. Thus, these locations are not homogeneously distributed in the latitude-longitude space and the data density in the latitude-longitude space close to the sub-satellite point is thus highest and decreases towords the edge of the Earth disk. Locations in this reduced geographical grid are selected randomly, taking care that all positions are considered. Atmospheric and surface properties are then collected according to their geographic locations.

### 3.1.2 Atmospheric profiles of gases and clouds

ECMWF IFS (Integrated Forecast System) analysis data is used for the majority of the quantities, i.e. surface pressure, geopotential, land-sea mask, skin temperature, temperature, specific humidity, ozone mass mixing ratio, cloud cover, cloud liquid water content (LWC), and cloud ice water content (IWC). This enables the compilation of realistic atmospheric profiles at the given locations of the simulation (Sect. 3.1.1). Data is defined on a latitude-longitude grid covering the entire globe. Vertical variability from the surface up to model top is considered with the full set of model levels. IFS data is available at different spatial and temporal resolutions depending on user needs and model version, thus the data characteristics selected for VADUGS are illustrated in Sect. 4.2 for the present application.

The 1D vertical atmosphere structure is set up by calculating pressure, temperature, height, humidity and ozone for each atmospheric layer from ECMWF data for the selected simulation location and time. The barometric formula is applied to compute the respective height range. Water vapour and ozone stem from ECMWF.

Atmospheric gas absorption properties are computed through the low resolution band models developed for the LOW-TRAN 7 atmospheric transmission code (Pierluissi and Peng, 1985) that uses an exponential sum fit with a resolution of $20\,\mathrm{cm}^{-1}$. We implemented 15 spectral grid points for each MSG2/SEVIRI thermal channel (Sect. 2.1) under consideration of the corresponding spectral response function. The code was adopted from SBDART (Ricchiazzi and Gautier, 1998).

Cloud vertical profiles of liquid and ice water content are obtained from IFS and effective radii necesary for the radiative transfer model (Sect.3.1.6) are parameterised. Based on these values, optical properties are then assigned to every cloud. This method is described in detail in Appendix A.

### 3.1.3 Atmospheric profiles of volcanic ash

Volcanic clouds as well as other airborne aerosols show variations in size, shape, and composition (see e.g. Langmann, 2013) as a consequence of differing origin conditions and transport processes like gravitational settling, wash-out, etc. In this study, volcanic ash is represented by a homogeneous layer with random values for its vertical position in the atmosphere, vertical extent, and homogeneous mass volume concentration. The assumption of airborne aerosol occurring in the form of one layer is common practice, see Lee et al. (e.g. 2014), although multiple layers are not uncommon in observations (e.g. Marenco et al., 2011; Schumann et al., 2011).

Top height is at most 18 km above the surface, vertical extent is limited to 2.5 km and volcanic ash bottom height is computed as top minus extent.

The refractive indices $m$ of volcanic ash are for Eyjafjallajökull as described in Appendix B. Please notice that this enables the retrieval to be tailored to this eruption, however the validation in Sect. 4.4.4 shows that VADUGS provides VAMC with a similar accuracy also for the Puyehue-Cordón Caulle eruption in 2011, thus indicating that its applicability could be extended to other volcanoes. Nevertheless, the usage of refractive indices for Eyjafjallajökull represents a principle limitation of VADUGS, that has been addressed by its successor (see Sect. 6).

### 3.1.4 Surface properties

The relevant surface property needed for realistic radiative transfer simulations in the infrared spectral range is the surface emissivity. According to the time selected for the ECMWF profiles (Sect. 3.1.2) the suitable monthly data from the surface emissivity data set is selected to ensure consistency. For land pixels it is obtained from Seemann et al. (2008) for the year 2010, where monthly surface emissivity maps are available globally at ten wavelengths located at 3.6, 4.3, 5.0, 5.8, 7.6, 8.3, 9.3, 10.8, 12.1, and 14.3 $\mu$m with 0.05° spatial resolution in latitude and longitude. The radiative transfer model interpolates linearly in between. For water pixels, a spectrally and timely constant value of 0.986 is used, corresponding e.g. to the sea surface emissivity measured in Niclòs et al. (2005) for wind speeds of 5–10 m/s in the spectral range 8–14 $\mu$m encompassing all relevant SEVIRI channels for a viewing zenith angle of 25°. The fact that monthly means are used for radiative transfer simulations on particular days at selected times of the day does not affect the quality of the resulting data set since it does not spoil the aim of producing realistic satellite observations.

### 3.1.5 Viewing geometry

The only relevant viewing geometry parameter in one dimensional thermal radiative transfer is the viewing zenith angle of the satellite (the position of the Sun is not relevant). Although we select fixed locations on the SEVIRI grid (Sect. 3.1.1), radiative transfer simulations are always run for a set of 41 cosines of the viewing zenith angles from 0.2 (corresponding to ~ 78° viewing zenith angle) to 1.0 (nadir). This is done for three reasons: (a) we do not want the NN to learn fixed relationships between latitude and viewing zenith angle (for this reason latitude is also no input variable of VADUGS, see Sect. 4.2); (b) the NN is given the chance to extract information about the transmissivity of the ash clouds as a function of viewing zenith angle from the input data; and (c) the resulting data set of simulated BTs will be filtered (Sect. 4.2) to keep only those that show a clear ash signal, which strongly reduces the number of ash loaded pixels available for training. This is meant to compensate for the fact that this approach might also lead to a more difficult learning procedure since not all meteorological conditions are observed for all viewing angles.

### 3.1.6 Radiative transfer model

The software package libRadtran (Emde et al., 2016; Mayer and Kylling, 2005) is used to realistically simulate satellite radiances and to compute results of satellite measurements in the form of BTs. All radiative transfer simulations make use of a C version of the 1D solver DISORT (Stamnes et al., 1988; Buras et al., 2011) with 16 streams. Input data relevant to the radiative transfer model in the thermal range have been described in the previous sections: satellite zenith angle, surface properties like temperature and surface emissivity, vertical profiles of temperature and gas concentrations, water and ice cloud properties like height, water content, and particle effective radius, as well as aerosol layer properties like height, aerosol optical properties, and mass concentration. Gas absorption is considered through LOWTRAN (see Sect. 3.1.2).

### 3.1.7 RTSIM

To manage and automate the generation of data sets, the program RTSIM was developed. RTSIM is coded in Python 3 and based on a corresponding module providing classes and functions as an interface to the underlying data base and the radiative transfer model. After start-up, all input data is accessed via its data management class. All environment parameters are prepared as input and suitable configuration commands are passed to the radiative transfer model. RTSIM supports independent processing modules, which can be used simultaneously, as well as communication through client applications during runtime. The integrity of defined functions is checked by calling a module-based self-test, which in turn performs several unit tests. RTSIM has been designed to run radiative transfer calculations in parallel. It combines multithreading with asynchronous handling of subprocesses in a scalable manner with respect to CPython's Global Interpreter Lock (GIL). For long-running subprocesses and a suitable number of available CPU cores, this can lead to significant decrease in computation time. RAM drives have been used to decrease I/O delays and thus decrease runtime. Results of radiative transfer simulations are stored in the ouput data set, together with all the information needed to determine the environment state including all mentioned quantities for surface and atmosphere.

At first, RTSIM selects randomly a day for the input atmospheric parameters (Sect. 3.1.2) and a SEVIRI grid point from the corresponding input file (Sect. 3.1.1). Its geographic coordinates are mapped by a nearest neighbour algorithm to the ECMWF grid such that skin temperature, gas and cloud profiles can be extracted (Sect. 3.1.2). Optical properties are then assigned to the cloud layers (Sect. 3.1.6). Surface properties for the pixel location are extracted from the suitable emissivity file (Sect. 3.1.4) and a volcanic ash layer (Sect. 3.1.3) is added to the atmosphere. Finally, the entire set of 41 satellite zenith angles described in Sect. 3.1.5 is assigned to the simulation.

The radiative transfer solver is called two times (with and without ash profile) when the atmosphere (Sect. 3.1.2) does not contain clouds. When the atmosphere contains cloud layers the radiative transfer solver is called four times (with and without ash profile for the atmosphere with and without clouds). In case ice clouds are present, ice crystal habit is selected randomly but is the same for the two calculations with clouds (with and without ash). Every call to the radiative transfer solver produces seven thermal SEVIRI observations for the 41 viewing geometries.

## 4 The VADUGS retrieval algorithm

This section describes the training data set used for the VADUGS implementation and the NN and its training.

### 4.1 The training data set

The training data set is compiled using the RTSIM tool (Sect. 3.1.7). To optimally cover different seasonal conditions we select day 15, 12 UTC for 12 months from February 2010 to January 2011. This period of time encompasses the Eyjafjallajökull eruption, for which the algorithm was initially created, and consists of data from one single ECMWF IFS model version (cycle CY36R1, started end of January 2010). Spatial resolution amounts to 0.25° in latitude and longitude, while the vertical grid

of atmospheric model levels encompasses 91 layers (ECMWF, 2010). Although only one time of the day is used, different local times can be covered also due to the fact that many viewing zenith angles are simulated for every atmospheric column. Furthermore, VADUGS only relies on thermal observations such that the position of the Sun above the horizon is not relevant. Nevertheless, nighttime variability of e.g. surface properties like temperature cannot be accounted for (this has been improved for VADUGS successor).

Volcanic ash layers are selected the following way. In 10,279,900 simulations ash mass volume concentration is a random number between 0.001 and $10\,\mathrm{g\,m^{-3}}$, while additional 1,200,000 simulations with mass volume concentration between 0.001 and $20\,\mathrm{g\,m^{-3}}$, 1,200,000 simulations with mass volume concentration between 0.05 and $2.15\,\mathrm{g\,m^{-3}}$ and 600,000 simulations with mass volume concentration between 0.001 and $1000\,\mathrm{g\,m^{-3}}$ have been performed to stress on one side medium range concentrations and on the other side also high concentration peaks. This makes a total of 13,279,900 atmospheric ash profiles used for the radiative transfer simulations.

Although the entire simulation data set provides a realistic and extensive basis for the training of a NN, three additional constraints have been implemented to make sure that a clear relationship between input and output data is provided to the NN. Thus, the results of the simulations have been filtered according to the following criteria:

$$BTD(10.8 - 12.0) \geq 0\,\mathrm{K} \tag{1}$$
$$BTD(8.7 - 10.8) \geq 0\,\mathrm{K} \tag{2}$$
$$BTD(8.7 - 12.0) \geq -5\,\mathrm{K}\,. \tag{3}$$

If one of these conditions is fulfilled, the corresponding VAMC and VATH are set to zero. The first one aims at identifying only pixels that are distinct from ice clouds, whose typical signature in split window channels is a negative $BTD(10.8-12.0)$ (e.g. Inoue, 1985), while the second and third condition shall exclude all low level liquid water clouds ($BTD(8.7-10.8) \geq 0\,K$ is typical for the ice phase, e.g. Baum et al., 2000). The training data set eventually contains 40,057,800 samples, 965,268 have non-zero VAMC and VATH, with a VAMC range from $0.12\,\mathrm{mg/m^2}$ to $2.446\,\mathrm{kg/m^2}$ and VATH from $22.2\,\mathrm{m}$ to $17.9\,\mathrm{km}$. The histogram of VAMC, of VATH and their combined histograms are given in Fig. 1. It shows that VAMC concentrate to values lower than $5\,\mathrm{g\,m^{-2}}$ and VATH extends to $14\,\mathrm{km}$ in the majority of the cases. Moreover, one can distinguish between weak eruptions (VATH up to $5\,\mathrm{km}$) and stronger eruptions with ash up to $14\,\mathrm{km}$. Although the data set before the filtering covers all combinations of VAMC and VATH, the filtered data set is reduced in this respect such that VAMC $> 6\text{-}7\,\mathrm{g\,m^{-2}}$ are not found at VATH larger than $7\,\mathrm{km}$ at most. This represents a limitation especially close to the vent of the volcano, but as seen below in the validation against CALIOP (Sect. 4.4.4) the values of VAMC found there are well below these values of $6\text{-}7\,\mathrm{g\,m^{-2}}$. Thus, we think that in most situations this does not represent a strong limitation for VADUGS.

The resulting brightness temperature differences, like BTD(8.7-12.0) and BTD(10.8-12.0) in Fig. 2, for VAMC $> 0\,\mathrm{g\,m^{-2}}$ show the expected behaviour, with negative values down to -25 K and a dependency on VAMC for VAMC smaller than approximately $5\text{–}6\,\mathrm{g\,m^{-2}}$. For higher VAMC, the BTD variations are much smaller, thus pointing at the physical limits of the passive thermal observations that reach saturation in this VAMC range. The dependency of brightness temperature at $10.8\,\mu m$ on VAMC is shown in the lowest panel in Fig. 2: BT(10.8) varies from very low values close to 200 K up to 320 K, thus indi-

cating that opaque ash layers are present at different heights. Large VAMC $> 5\,\mathrm{g\,m^{-2}}$ correspond to brightness temperatures between 260 and 280 K, thus corresponding to medium height levels up to approximately 5 km (see Fig. 1).

## 4.2 Training the neural network

The simulated observations presented above are used as input for the training of a NN. As input variables for the NN the brightness temperatures from all seven thermal SEVIRI channels are selected together with a set of brightness temperature differences: BTD(8.7-9.7), BTD(8.7-10.8), BTD(8.7-12.0), BTD(8.7-13.4), BTD(9.7-12.0), BTD(9.7-13.4) and BTD(6.2-7.3). The BTDs containing window channels alone (8.7, 10.8 and 12.0 $\mu$m channels) are meant to yield the physics of thin layers in the atmosphere. Of course, other BTDs, like the most used BTD(10.8-12), can be implicitly obtained by the NN through combination of the available ones.

BTDs with the $CO_2$ channel (centred at 13.4 $\mu$m) are supposed to transport direct information about ash layer height, since the vertical weigthing function of this channel has a broad peak in the troposphere (and for this reason is used for $CO_2$ slicing, see e.g. Menzel et al., 1983). Finally, BTDs with the ozone channel (centred at 9.7 $\mu$m) carry information about the ozone yearly cycle and its geographic distribution. Due to large possible variations in stratospheric ozone columns during the course of the year and to the considerable variability in surface temperatures, especially at high latitudes, this channel must be handled with care in order to prevent misdetections, as was for instance the case in (Ewald et al., 2013) with respect to cirrus cloud detection. Although the NN is able to deduce relationships between input variables during training, the provision of brightness temperatures differences is supposed to facilitate and speed up the learning process since these quantities explicitly contain the physics of the problem. However, the usage of a NN prevents us from defining a priori thresholds to the BTDs like e.g. in Yu et al. (2002) and Francis et al. (2012). Furthermore, skin temperature (from ECMWF, Sect. 3.1.2) together with a land/sea flag are selected as input in order to describe surface emission properties. Surface emissivity is neglected here since its variability is thought to be of secondary importance compared to surface temperature and because it is difficult to obtain daily/hourly values of this quantity, especially for a possible near real-time application. Finally, viewing zenith angle helps considering the slant path of radiation through the atmosphere. This makes a total of 17 input variables.

Output variables are, as already mentioned, VAMC and VATH (two output variables), the topology of the VADUGS NN and the backpropagation procedure have been presented in Sect. 2.2. The final training data set described above has been randomly mixed and fed into the NN.

## 4.3 Applying the neural network

For the application of the NN, BTs from all thermal channels are needed according to the previous section, but also a land/sea flag, that we consider fixed in time, and a viewing zenith angle map, that can also be considered as fixed if the spacecraft is located above 0° E over the equator and fluctuations around this point are neglected. Nevertheless, skin temperature must be provided still. This quantity has been extracted from ECMWF analysis data for the training (Sect. 3.1.2), but it might be in principle obtained from other sources as well. A short discussion about this aspect is given in Sect. 5.

The output quantities are VAMC and VATH. The detection of ash contaminated pixels must be performed through VAMC and is discussed in detail in the next section. Here we show a sequence of false colour RGB pictures with VAMC overlays from 13 May 2010 04 UTC to 17 May 2010 16 UTC in 12 h steps (Fig. 3) during the third phase of the Eyjafjallajökull eruption (Langmann et al., 2012). The false colour RGBs are based on the SEVIRI2 solar channels centred at $0.6\,\mu$m, $0.8\,\mu$m and the inverted thermal $10.8\,\mu$m channel. During nighttime, only temperature information is available such that blue shades are obtained, during daytime colours mimick true colour images. In all panels, black areas correspond to ice clouds as detected by the COCS algorithm (Kox et al., 2014): here no assertion about the presence of ash can be made. Red colours indicate VAMC from 0 to $3\,\mathrm{g\,m^{-2}}$ (see scale at the bottom), pixels with VAMC values lower than $0.05\,\mathrm{g\,m^{-2}}$ are treated as ash free. The region selected comprises Iceland in the North-West corner, Great Britain in the centre, Scandinavia in the North-East part and France-Germany-Denmark over central Europe. The VADUGS retrieval clearly shows the ash emitted by the Eyjafjallajökull volcano in the South of Iceland and how the ash cloud is transported by winds first towards the East (13 May 2010 04 UTC), then towards South-East. Through the use of thermal observations, ash information can be derived with the same accuracy during both day and night. The cloud reaches Great Britain on 14 May 2010 and is observable there for many hours, while the winds close to the volcano change and blow ash to the West along high latitudes. From 16 May 2010 the ash is carried again towards Great Britain, with some small patches reaching the continent. On 17 May 2010 16 UTC a relatively large ash cloud is located above the North Sea, where it has been probed by both the UK FAAM aircraft (Marenco et al., 2011) and the German DLR Falcon (Schumann et al., 2011). In general, dark red colours are typical for the ash plume close to the vent indicating high ash concentrations, while the red colours become fainter further away from the volcano, as expected through sedimentation and/or interactions with water clouds. From time to time single ash contaminated pixels are found: in many cases, through the visual evaluation of the full 15 min temporal evolution of the ash, they can be ascribed to thin diluted ash cloud patches, in some cases they seem to be probably false alarms. Thus, the algorithm is able to detect the Eyjafjallajökull ash cloud for which it has been developed in a very plausible way.

## 4.4 Validation

Two approaches are selected for validation of the NN. First, the NN's performance is evaluated against simulated observations to assess how well the retrieval has learnt the relationships between input and output variables contained in the training data set. The second approach consists in the evaluation of the NN against CALIOP observations of volcanic ash to assess the quality of the retrieval in real situations. The metrics used for this are presented in Appendix C.

### 4.4.1 Simulated validation data set

A second data set is simulated along the lines of the training data set (Sect. 4.1). Since most of the input data samples are concentrated to VAMC$\leq5\,\mathrm{mg\,m^{-3}}$ and VATH$\leq14\,$km, these limits are used for the compilation of the validation data set. Of course, the same ash optical properties are used and the ash layer extent reaches up to 2.5 km. Again, various mass concentration regimes are selected to compose the ash profiles: $0\,\mathrm{mg\,m^{-3}}$, 0.001 to $0.1\,\mathrm{mg\,m^{-3}}$, 0.1 to $0.2\,\mathrm{mg\,m^{-3}}$, 0.15 to $0.25\,\mathrm{mg\,m^{-3}}$, 0.2 to $0.3\,\mathrm{mg\,m^{-3}}$, 0.3 to $1\,\mathrm{mg\,m^{-3}}$, 1 to $2\,\mathrm{mg\,m^{-3}}$, 1.5 to $2.5\,\mathrm{mg\,m^{-3}}$, 2 to $3\,\mathrm{mg\,m^{-3}}$ and 3 to $10\,\mathrm{mg\,m^{-3}}$. For each ash

profile, samples with and without volcanic ash/meteorological clouds are simulated in various height ranges (tops between 0.5 and 14 km), thus producing the gaps observed in Fig. 4. After application of the filter implemented in Sect. 4.1 with Eqs. 1, the data set comprises 3,526,397 samples with 100,083 ash loaded samples (those plotted in Fig. 4). Most points accumulate to concentrations below $1\,\mathrm{g\,m^{-2}}$ and only few samples at higher concentrations.

### 4.4.2 Ash detection

The detection of volcanic ash clouds is performed by applying a threshold to the mass load retrieval.

Figure 5 shows the POD against the FAR for different thresholds (indicated by the color bar). Considering all retrieval outputs with VAMC $> 0\,\mathrm{g\,m^{-2}}$ results in a POD of 0.96 and a FAR of 0.36. Increasing the threshold value decreases both the POD and FAR. At first, the FAR decreases faster than the POD and close to a threshold value of $0.1\,\mathrm{g\,m^{-2}}$ the POD starts to decrease faster as well. Thus, in the application of the VADUGS retrieval the threshold value of $0.1\,\mathrm{g\,m^{-2}}$ is selected as a trade-off between high POD and low FAR: here, averaged over the entire validation data set, the POD amounts to 0.92 and the FAR to 0.08. For the threshold of $0.05\,\mathrm{g\,m^{-2}}$ used in Sects. 4.3 and 4.4.4 POD amounts to 0.95 and FAR to 0.17.

Although the values of POD and FAR for a threshold of $0.1\,\mathrm{g\,m^{-2}}$ are very good, it has to be noticed that the validation data set is not well balanced with respect to ash loaded and ash free samples, since the latter make up a very large fraction of them. In addition, many ash free samples also show BTD(10.8-12.0) values close to 0 K. While negative BTDs are thought to stem mainly from ash clouds, some of them also contain meteorological clouds. Thus, to further simplify the task of the NN we introduce an additional a-priori filter:

$$BTD(10.8\text{-}12.0) > -0.6\,\mathrm{K}\,. \tag{4}$$

Pixels satisfying this empirical condition are set to VAMC=$0\,\mathrm{g\,m^{-2}}$ and VATH is undefined. This condition applies to 2,789,235 samples of the validation data set, 2,777,879 of them being already ash free and 11,356 ash loaded, with ash column concentrations ranging from 0 to $5\,\mathrm{g\,m^{-2}}$. This ensures that 81.1 % of the ash free samples are already correctly classified before the application of VADUGS, at the cost of 11.3 % of the ash loaded samples being missed. With the help of this additional filtering, the overall POD sinks to 0.84 but the FAR also sinks to 0.05. Even if the gain in FAR (FAR measures how large fraction of the ash free points are falsely classified as being ash, see Appendix C) is low, this can result in many pixels being misclassified as ash, which would artificially enlarge the area covered by ash that should be avoided by air traffic and is thus preferred to a larger POD. This filtering is always applied in the rest of the manuscript, and in all other applications as for instance at DWD (Sect. 5).

Finally, considering the POD as a function of true VAMC in Fig. 6 shows that VADUGS, even if the concentration threshold of $0.1\,\mathrm{g\,m^{-2}}$ is applied, is able to detect 60 to 70 % of the ash samples with true VAMC$\leq 0.1\,\mathrm{g\,m^{-2}}$. POD increases with VAMC and for VAMC $= 0.5\,\mathrm{g\,m^{-2}}$ it already reaches its maximum around 90 %. On the other hand, a POD of 100 % is never reached, even for large VAMC, both because these cases might resemble thick (ash/meteorological) clouds or because they might be affected by the presence of water/ice clouds.

### 4.4.3 Validation of ash concentration and height against simulated observations

For the validation of VAMC and VATH we consider only ash-laden samples as a function of $\mu = \cos\vartheta$, the cosine of the satellite viewing zenith angle. Six subsets are defined, with all samples included in subset #0, samples with $0 < \mu \leq 0.2$ ($90° > \vartheta \geq 78.5°$) in subset #1, $0.2 < \mu \leq 0.4$ ($78.5° > \vartheta \geq 66.4°$) in subset #2, $0.4 < \mu \leq 0.6$ ($66.4° > \vartheta \geq 53.1°$) in subset #3, $0.6 < \mu \leq 0.8$ ($53.1° > \vartheta \geq 36.9°$) in subset #4 and $0.8 < \mu \leq 1.0$ ($36.9° > \vartheta \geq 0°$) in subset #5. Results of this validation are summarised in Tab. 1 for three subgroups: where true VAMC is a) smaller than $0.5\,\mathrm{g\,m^{-2}}$ (only thin ash layers) or b)

smaller than $2.0\,\mathrm{g\,m^{-2}}$ (thin and thick ash layers) or c) smaller than $5.0\,\mathrm{g\,m^{-2}}$ (all simulated ash layers). The values for the last subgroup represent the full validation data set, which is first discussed in the next lines. Apart from the fact that the subsets contain different numbers of samples, as indicated by $N$, it is obvious that VADUGS results scatter considerably. The correlation between VADUGS and the simulated truth is not strong, with Pearson coefficients below 0.3. The best correlation is achieved for subset #3 ($0.4 < \mu \leq 0.6, 66.4° > \vartheta \geq 53.1°$), with also the lowest MAPE and MPE of 72 % and -2 % respectively.

For subset #4 ($0.6 < \mu \leq 0.8, 53.1° > \vartheta \geq 36.9°$) MAPE and MPE are slightly larger (113 % and 27 % respectively). For very high viewing zenith angles (subset #1), deviations are very large (both MAPE and MPE > 200 %), while small viewing zenith angles produce a MAPE of 180 % and MPE of 104 %. Thus, VADUGS works best for moderate viewing zenith angles but always struggles with the determination of the correct VAMC, usually leading to an overestimation. Considering the first two subgroups (VAMC < $0.5\,\mathrm{g\,m^{-2}}$ and VAMC < $2.0\,\mathrm{g\,m^{-2}}$), we see that for thin ash (VAMC < $0.5\,\mathrm{g\,m^{-2}}$) uncertainties (MAPE,

MPE) are largest, indicating that the determination of VAMC for thin ash layers is most difficult. In general, MAPE and MPE for VAMC < $2.0\,\mathrm{g\,m^{-2}}$ and VAMC < $5.0\,\mathrm{g\,m^{-2}}$ are comparable, with slightly better results for the latter. However, RMSE are (much) lower (between 0.39 and $0.58\,\mathrm{g\,m^{-2}}$) and the correlation coefficients are higher (up to 0.39) for VAMC < $2.0\,\mathrm{g\,m^{-2}}$, meaning that the most reliable results are for ash columns larger than $0.5\,\mathrm{g\,m^{-2}}$ and smaller than $2.0\,\mathrm{g\,m^{-2}}$. Again, subsets #3 and #4 are the ones with the lowest MAPE and MPE not only for the full data set but also for the other two VAMC subgroups.

In subset #3 in particular, MPE is very close to zero (+5 % for VAMC< $2.0\,\mathrm{g\,m^{-2}}$).

For VATH, validation results are collected in Tab. 2 where the dependency of the VADUGS errors on true VATH for three intervals – true VATH < 8 km, 12 km and 14 km – is described. Again, we start the discussion with the last subgroup (VATH < 14 km) that contains the full validation data set. The general features correspond to those for VAMC. In particular, the large scatter of VADUGS values is evident, although correlation coefficients are higher for VATH (up to 0.54) than for VAMC.

Similarly, MAPE and MPE values are lower than for VAMC: considering all validation data, MAPE amounts to 54 % with an underestimation (MPE) by -34 %. Although in general VATH is underestimated by VADUGS (negative MPE), this effect is weakest (-11 %) for small viewing zenith angles (subset #5) but connected with a relatively high MAPE of 65 %. The smallest MAPE is for subset #1 (44 %), i.e. for large viewing zenith angles, where RMSE is also smallest (4.73 km) yet high. For subsets #2–#4 absolute error and underestimation are high (approximately 60 % and -40 %), with RMSE around 6 km, i.e. VADUGS

provides a moderate sensitivity to VATH in these situations.

Comparing the three VATH intervals in Tab. 2 with each other, one can see that, apart from the dependency on viewing zenith angle, the MAPE and MPE for all three VATH intervals are very similar for subsets #1 and #2. However, for subsets #2 to #5, in

**Table 1.** Statistical evaluation of VAMC from VADUGS against simulated observations (representing the truth).

| Viewing angle subset | true VAMC upper bound g m$^{-2}$ | N | Pearson | MAPE % | MPE % | RMSE g m$^{-2}$ |
|---|---|---|---|---|---|---|
| all | 0.5 | 50424 | 0.20 | 258 | +241 | 0.38 |
| all | 2.0 | 76461 | 0.26 | 188 | +148 | 0.50 |
| all | 5.0 | 84285 | 0.17 | 178 | +127 | 0.91 |
| # 1 | 0.5 | 24867 | 0.25 | 322 | +314 | 0.49 |
| # 1 | 2.0 | 33463 | 0.29 | 252 | +231 | 0.52 |
| # 1 | 5.0 | 36117 | 0.20 | 239 | +209 | 0.78 |
| # 2 | 0.5 | 12420 | 0.33 | 205 | +182 | 0.23 |
| # 2 | 2.0 | 17735 | 0.36 | 160 | +117 | 0.39 |
| # 2 | 5.0 | 18929 | 0.23 | 155 | +105 | 0.80 |
| # 3 | 0.5 | 5140 | 0.40 | 82 | +51 | 0.15 |
| # 3 | 2.0 | 9730 | 0.39 | 71 | +5 | 0.50 |
| # 3 | 5.0 | 10602 | 0.27 | 72 | -2 | 0.87 |
| # 4 | 0.5 | 3780 | 0.18 | 178 | +149 | 0.20 |
| # 4 | 2.0 | 7781 | 0.33 | 118 | +46 | 0.58 |
| # 4 | 5.0 | 9140 | 0.23 | 113 | +27 | 1.17 |
| # 5 | 0.5 | 4217 | 0.11 | 323 | +306 | 0.30 |
| # 5 | 2.0 | 7752 | 0.28 | 203 | +146 | 0.57 |
| # 5 | 5.0 | 9497 | 0.20 | 180 | +104 | 1.27 |

all three intervals underestimation of VATH through VADUGS becomes less pronounced with decreasing viewing zenith angle, i.e. when going from subset #2 to #5, with lowest underestimations for low ash clouds (VATH < 8 km). In subgroup #2 MPE for VATH < 8 km amounts to -42 %, to -25 % for subset #3, -9 % for subset #4 and +17 % for subset #5. In the three subsets #3–#5 correlation is highest for the low VATH regime (Pearson coefficients between 0.28 and 0.33). Nevertheless, MAPE is always close to 60 % and reaches 70 % for subset #5.

Since thin ash clouds are difficult to detect and the determination of their properties is complex, we investigate the accuracy of VATH when VAMC is larger than 0.3, 0.9 and 2.0 g m$^{-2}$ and VATH is below 12 km in Tab. 3. In almost all cases, Pearson correlation coefficients increase with increasing VAMC threshold, i.e. it is easier for VADUGS to assess VATH when the ash cloud is thick. For VAMC > 2.0 g m$^{-2}$, correlation coefficients larger than 0.7 can be achieved. At the same time, MAPE lies between 46 and 59 % and MPE between -44 and +23 % for the clouds with VAMC > 2.0 g m$^{-2}$. Furthermore, underestimation of VATH in subsets #1–#3 turns into an overestimation in subsets #4 and #5. However, for all viewing angle subsets and all VATH subgroups MAPE is always relatively high between 40 and 65 %.

**Table 2.** Statistical evaluation of VATH from VADUGS against simulated observations (representing the truth).

| Viewing angle subset | true VATH upper bound km | N | Pearson | MAPE % | MPE % | RMSE km |
|---|---|---|---|---|---|---|
| all | 8.0 | 32149 | 0.36 | 58 | -24 | 3.43 |
| all | 12.0 | 74780 | 0.40 | 54 | -33 | 5.04 |
| all | 14.0 | 84285 | 0.38 | 54 | -34 | 5.49 |
| # 1 | 8.0 | 11305 | 0.52 | 50 | -35 | 3.00 |
| # 1 | 12.0 | 31334 | 0.55 | 45 | -31 | 4.34 |
| # 1 | 14.0 | 36117 | 0.54 | 44 | -31 | 4.73 |
| # 2 | 8.0 | 7448 | 0.38 | 60 | -42 | 3.45 |
| # 2 | 12.0 | 17029 | 0.40 | 58 | -48 | 5.39 |
| # 2 | 14.0 | 18929 | 0.36 | 59 | -49 | 5.91 |
| # 3 | 8.0 | 4077 | 0.31 | 59 | -25 | 3.69 |
| # 3 | 12.0 | 9501 | 0.24 | 61 | -41 | 5.76 |
| # 3 | 14.0 | 10602 | 0.21 | 62 | -44 | 6.27 |
| # 4 | 8.0 | 4372 | 0.33 | 59 | -9 | 3.64 |
| # 4 | 12.0 | 8285 | 0.24 | 62 | -27 | 5.54 |
| # 4 | 14.0 | 9140 | 0.21 | 62 | -31 | 6.05 |
| # 5 | 8.0 | 4947 | 0.28 | 70 | +17 | 3.86 |
| # 5 | 12.0 | 8631 | 0.24 | 65 | -6 | 5.29 |
| # 5 | 14.0 | 9497 | 0.20 | 65 | -11 | 5.79 |

Generally, VADUGS struggles with the determination of the correct mass load and top height when applying it to observations filtered according to Eqs. 1. However, with a more stringent filtering that identifies simulated ash contaminated observations within the validation data set fulfilling

$$BTD(10.8 - 12.0) < -1 \tag{5}$$

$$BTD(8.7 - 10.8) < -1 \tag{6}$$

$$BTD(8.7 - 12.0) < -6 \tag{7}$$

(that is meant to isolate cases where the ash signal is stronger in the simulated observations) and considering the findings above, VADUGS performs significantly better. Figure 7 shows scatter plots between true and retrieved VAMC for true VAMC in the range $0.0$–$2.0\,\mathrm{g}\,m^{-2}$ (see Tab. 1) for this reduced validation data set. Pearson coefficients are clearly higher than with the previous filtering, with values up to 0.51, but overestimation remains and leads to high MAPE and MPE, with again subset

**Table 3.** Statistical evaluation of VATH from VADUGS against simulated observations (representing the truth) for VA with true VATH $\leq$ 12 km.

| Viewing angle subset | true VAMC lower bound $\mathrm{g\,m^{-2}}$ | N | Pearson | MAPE % | MPE % | RMSE km |
|---|---|---|---|---|---|---|
| all | 0.3 | 46673 | 0.39 | 50 | -22 | 4.78 |
| all | 0.9 | 19138 | 0.38 | 53 | -11 | 4.82 |
| all | 2.0 | 7150 | 0.39 | 51 | -14 | 4.33 |
| # 1 | 0.3 | 16863 | 0.58 | 39 | -22 | 3.85 |
| # 1 | 0.9 | 5864 | 0.60 | 41 | -30 | 4.03 |
| # 1 | 2.0 | 2241 | 0.72 | 48 | -44 | 4.46 |
| # 2 | 0.3 | 10384 | 0.42 | 50 | -33 | 4.78 |
| # 2 | 0.9 | 3203 | 0.49 | 49 | -20 | 4.19 |
| # 2 | 2.0 | 1128 | 0.64 | 46 | -40 | 3.81 |
| # 3 | 0.3 | 7048 | 0.24 | 57 | -30 | 5.46 |
| # 3 | 0.9 | 2966 | 0.34 | 57 | -6 | 5.08 |
| # 3 | 2.0 | 834 | 0.26 | 46 | -7 | 4.05 |
| # 4 | 0.3 | 6267 | 0.23 | 60 | -19 | 5.55 |
| # 4 | 0.9 | 3463 | 0.30 | 61 | 0 | 5.43 |
| # 4 | 2.0 | 1293 | 0.34 | 52 | +10 | 4.25 |
| # 5 | 0.3 | 6111 | 0.26 | 64 | +1 | 5.38 |
| # 5 | 0.9 | 3642 | 0.30 | 65 | +15 | 5.63 |
| # 5 | 2.0 | 1654 | 0.35 | 59 | +23 | 4.69 |

#3 having the smallest errors (MAPE=72 %, MPE= 25 %). However, RMSE is in almost all cases smaller than in the general evaluation.

Fig. 8 shows scatter plots for the cloud top height retrieval when VAMC $> 0.3\,\mathrm{g}\,m^{-2}$. Generally, VADUGS again underestimates top height, but to a smaller degree than with the previous filtering. At the same time, correlations coefficients reach up to 0.74 and RMSE is usually smaller. Best results are obtained for subset #1, i.e. for the smallest cosine viewing zenith angles.

### 4.4.4 Validation of ash concentration and height against CALIOP observations

To validate VADUGS under real conditions, CALIPSO (Cloud-Aerosol Lidar and Infrared Pathfinder Satellite Observation, Winker et al., 2009) version 4.10 level 2 aerosol products including data on stratospheric ash layers (Kim et al., 2018) are used. They are obtained from CALIOP (Cloud-Aerosol Lidar with Orthogonal Polarization) retrievals. The final spatial resolution after processing is 5 km horizontally and 60 m vertically. Samples are discarded if the extinction quality control flag (qc) is

**Table 4.** CALIOP measurements used for the statistical evaluation; all CALIPSO flyovers took place in June 2011.

| Time / UTC | Lat. / ° | Lon. / ° | Samples |
|---|---|---|---|
| 15th, 1830-1840 | -59.6 to -40.4 | -57.5 to -66.0 | 300 |
| 16th, 1551-1605 | -48.8 to -39.3 | -24.4 to -27.7 | 162 |
| 16th, 1729-1743 | -60.9 to -44.6 | -42.7 to -50.7 | 187 |
| 17th, 0300-0313 | -40.6 to -62.0 | -27.7 to -37.8 | 251 |
| 17th, 1455-1510 | -44.4 to -37.3 | -12.1 to -14.4 | 82 |
| 18th, 0204-0218 | -35.4 to -64.4 | -12.2 to -25.9 | 199 |
| selected subset | | | 1081 |
| (VAMC $> 0.05\,\mathrm{g\,m^{-2}}$ for CALIOP/VADUGS) | | | |

neither one (indicating that the lidar ratio is directly retrieved from the data) nor zero (initial lidar ratio leads to stable extinction retrievals). For the remaining samples the optical depth at 532 nm due to volcanic ash is derived from the extinction profile using only the samples classified as ash. For samples with qc=0, a correction is made to take care of the difference between the default lidar ratio of ash (44 sr) used in version 4.10 and that obtained from the direct (qc=1) retrievals (58 sr). The mass loading is estimated from the optical depth using a mass extinction coefficient of $0.69\,\mathrm{m^2 g\,m^{-1}}$ (Gasteiger et al., 2011; Winker et al., 2012), whereas the highest ash layer determines the ash cloud height.

For the comparison, the Puyehue-Cordón Caulle eruption in 2011 is considered: six daytime and nighttime orbits of CALIPSO over ash clouds in the southern Atlantic between 15 and 18 June 2011 are collected (Tab. 4). VAMC range between $0\,\mathrm{g\,m^{-2}}$ and $1.7\,\mathrm{g\,m^{-2}}$ and VATH between 10 and 15 km. Fig. 9 shows an example with the CALIPSO data (blue) plotted alongside the corresponding VADUGS retrievals (red); VAMC from CALIPSO has an uncertainty of a factor ~2 (light blue). Note that this reference data set is rather limited in the number of samples as well as in the variability of the observations, which are only of a single eruption, only above sea, can be assumed to have rather similar atmospheric conditions, and with ash layers only above and around the tropopause. Nevertheless, this eruption represents a good testbed for VADUGS that has been developed for Eyjafjallajökull.

The CALIOP retrievals are compared to VADUGS results for the temporally closest SEVIRI2 image. A parallax correction is applied to the latter such that the top of the ash layer is observed.

The example in Fig. 9 shows that the VAMC retrieval is generally in agreement considering the uncertainty of the CALIOP data; a tendency of VADUGS towards overestimations is visible, as discussed in the previous section. The cloud top derived by VADUGS is lower than the reference values when the ash layer is thin. A reasonable VATH retrieval is performed at one of the thicker parts of the observed ash cloud (at a latitude around -48°). This behaviour basically confirms the analysis in the previous section based on simulated satellite observations.

Only samples with VAMC retrieval larger than $0.05\,\mathrm{g\,m^{-2}}$ using both CALIOP and VADUGS are considered (i.e. the threshold on $0.1\,\mathrm{g\,m^{-2}}$ used before has been lowered in order to increase the number of ash conatminated pixels). The scatter

plot Fig. 10 compares the VAMC of the two retrievals. The measurement points are located around the identity, also expressed by a Pearson coefficient of 0.49, which is comparable to the correlations coefficients obtained in the analysis in the previous section. However, once again VADUGS tends to overestimate VAMC as can be seen from the positive MPE of 55%. Generally, the MAPE is 90%. Note that MAPE, MPE and Pearson coefficient are close to the values in Tab. 1 for subset #3 with an upper VAMC threshold of $0.5\,\mathrm{g\,m^{-2}}$.

The histogram in Fig. 11 shows the VATH distribution retrieved from CALIOP and VADUGS. The CALIOP measurements peak around 12 to 13 km. The distribution of VADUGS-retrieved VATH peaks at 0 km and has a flank reaching up to 19 km, with a notable minor peak at about 9 to 12 km. 86 % of the retrieved VATH are < 8 km, and only 14 % are larger. This underlines the results of the example (Fig. 9) that VADUGS is able to retrieve the correct VATH for thick ash clouds, but generally underestimates it. However, one has also to consider here the fact that VATH determination above or close to the tropopause is particularly challenging due to the fact that the atmospheric temperature profile can be constant or increase with height, while below the tropopause, where VADUGS has been mainly trained for, temperature decreases with height. Notice that the general overestimation of VAMC is again related to (or maybe induced by) the underestimation of VATH.

### 4.4.5 Comparison to a non-machine learning based volcanic ash retrieval

VADUGS is a machine learning based algorithm for the detection of volcanic ash and the derivation of its properties. Initially, volcanic ash detection was based on brightness temperature difference tests using fixed thresholds for all meteorological conditions. The use of neural networks is intended to provide more flexibility in the application of the volcanic ash retrieval such that it is expected to easily adapt to the given atmospheric conditions (low level clouds, water vapour, land/sea...). The goal of this section is to provide a qualitative comparison of VADUGS to a "standard" retrieval of volcanic ash to check this hypothesis. Among the retrievals available in the literature, we have selected the 3-bands method (Guéhenneux et al., 2015) as it represents a strong improvement with respect to the 2-bands method first implemented by Prata (1989a,b) and because it is made available to the public through the web via the HOTVOLC real-time monitoring service (Gouhier et al., 2016) developed and managed by the Observatoire de Physique du Globe de Clermont-Ferrand (OPGC). Furthermore, it is easy to implement and has been shown to provide the main feature of volcanic ash clouds although its improved version, the 5-bands method (Gouhier et al., 2020), provides both an improved POD and reduced FAR.

For the comparison we have selected four different scenes from the second phase of the Eyjafjallajökull eruption. The first scene is taken from Guéhenneux et al. (2015) (their Fig. 12), where on 10 May 2010 00 UTC volcanic ash is seen to drift from the vent of the volcano towards the West. In Fig. 12 the 3-bands result for this time nicely shows in articular ash freshly emitted by the volcano and blown by the wind towards the South as well as other ash spots that are heading towards Greenland. One can notice that the 3bands method seems to produce some scattered results inside ice clouds. While some of them are close to Iceland and might represent real ash contaminated pixels, other (like those West of France) are probably false alarms. With the use of an ice mask, these pixels can easily be removed and do not represent a problem. The corresponding result from VADUGS (top right panel with the label 201005100000) misses the ash close to the vent and spots only few pixels West of Ireland. Instead, VADUGS observes more ash pixels further in the West close to -30°E +50°N, where also 3-bands observes

an ash cloud. But VADUGS also retrieves more ash contaminated pixels along the Greenland coast, an additional ash cloud around -20°E +55°N and scattered ash pixels between 50° and 60°N. Although it is not possible to assess which pixels are correctly classified as ash by the two retrievals without additional information, the temporal evolution of the ash distribution (not shown) indicates that the presence of ash in all these regions further away from the volcano is plausible and follows the general circulation patterns during the previous days.

The second day we have selected is 13 May 2010, for which we present two slots 04 and 16 UTC (second and third row in Fig. 12), as in Fig. 3.In the early morning of 13 May (second row in Fig. 12), ash emitted from Eyjafjallajökull is blown towards the East till the Faroe Islands, where the ash clouds is then diverted to the North. Apart again from some "noise" in correspondece with ice clouds, the 3-bands method detects again very nicely the ash plume between volcano and Faroe Islands, and also an ash cloud North of them. In comparison, VADUGS detects only a very small stripe of ash close to the vent, but then retrieves larger ash clouds close to the Faroe Islands and North of them. All these detections, for both 3-bands and VADUGS, are plausible and correspond to "smoke-like" structures in the RGB. In the afternoon (third row in Fig. 12), ash extends again from the volcano to the Faroe Islands, but it is first blown into the South-East direction (down to 60°N) and the to the North-East, bevore in correspondence with the Faroe Islands it is again driven to the North and the back to the West. This time, it is the the 3-bands method that partly misses the ash plume close to the vent, while VADUGS observes an almost completely connected area of ash contaminated pixels from Iceland to the Faroe, and a larger ash region North of the Faroe Islands.

Finally, in the last row of Fig. 12, we show the ash distribution over Europe on 17 May 2010 at 16 UTC. In this case, the 3-bands method does not provide almost any ash pixels while VADUGS observes large regions of ash over the North Atlantic between approx. -10° and +5°E and +58° and +68°N. The presence of ash here is plausible according e.g. to model simulations (see e.g. Fig 10f in Plu et al., 2021). Furthermore, VADUGS retrieves two ash clouds over the North Sea, where on that day it has been probed both by the UK FAAM aircraft (Marenco et al., 2011) and the DLR Falcon aircraft (Schumann et al., 2011).

In summary, both retrievals provide valuable results for the monitoring of volcanic ash, each one with its own strengths and deficiencies. While the 3-bands method seems to be able to detect ash clouds very close to the vent, where ash optical thickness is particularly high, in general VADUGS detects more ash further away from the volcano where optical thickness is assumed to be lower. However, exceptions exist – for instance the ash detected by the 3-bands method over the Atlantic South of Greenland on 10 May 2010 at 00 UTC or the ash directly observed to leave the volcano by VADUGS on 13 May 2010 at 16 UTC – and show that ash detection by passive sensors is a great challenge. In this sense, every ash cloud retrieval represents an additional piece of information that could and should be used in the context of early warning systems for aviation (e.g., Brenot et al., 2021).

## 5 Implementation at DWD

VADUGS has been implemented at DWD and runs operationally 24/7. Like in the example in Fig. 3 the ice cloud algorithm COCS (Kox et al., 2014) has been first implemented to identify ice cloud pixels that may shade the ash from the satellite's

view. This algorithm is currently being replaced by its successor (Strandgren et al., 2017). VADUGS and COCS are driven by a shell script, handling the input and output data and the call of the algorithms. The retrieval results are rasterised and saved as cf conform netcdf. The data transfer rate to the cockpits of aeroplanes is limited. Thus, the raster data is polygonised in order to keep the essential information but to reduce the size of the data to be transferred to the cockpit. The VADUGS results can be also visualised at the meteorological workplace of DWD and are therefore available for the forecasters at the

central prediction centre and the regional aviation weather advice centres of DWD. ecFlow is used for the operational 24/7 call and monitoring of the job. Ecflow is a workflow package which has been developed to run a large number of programs in a controlled environment, providing restart and monitoring capabilities (via web page or email). It is used at DWD to run all operational suites on the high performance computer. ecFlow is developed and maintained by ECMWF.

VADUGS has been trained with surface temperature from the IFS model operated by ECMWF (Sect.3.1.2). However, for

the 24/7 operation at DWD surface temperatures from the DWD weather model ICON (ICOsahedral Non-hydrostatic Zängl et al., 2015) are used. Every model has its own spatial bias characteristics, depending beside other factors on the surface type. The operational verification at DWD indicates that differences in the surface temperature could be in the order of 1–2 K. Thus, a sensitivity study was performed in order to estimate the effect of uncertainties in surface temperature on the ash concentration and top height derived from VADUGS when surface temperatures increased by +1 K or +2 K. Observations on 10 May 2010

during the Eyjafjallajökull eruption in the triangle enclosed by the Faroe Islands in the South, the Shetland Islands in the West and the northernmost tip of Great Britain were reprocessed for this purpose (Fig. 13, top). This day is characterized by a thick ash plume over the North Atlantic ocean moving from the North-West to the South-East with small concentrations over the land surfaces. The differences in VAMC between the normal run and the run with surface temperatures increased by +1 K and +2 K are about $0.015\,\mathrm{g\,m^{-2}}$ (<1%) and $0.03\,\mathrm{g\,m^{-2}}$ (<3%) for a +1 K and +2 K increase in surface temperature (Fig. 13, centre and

bottom). Considering the retrieval uncertainty (Sects. 4.4.3 and 4.4.4) these values are small. Thus, using ICON data should pose no significant problem to the ash plume retrieval. However, close to the ash plume there are several isolated pixel clusters with differences up to $0.15\,\mathrm{g\,m^{-2}}$. The increase in surface temperature leads to a detection (or misdetection) of ash for pixels where the column concentration was close but still below the detection limit of $0.1\,\mathrm{g\,m^{-2}}$ (Sect. 4.4.2). The top height shows no such effect (not shown): differences in top height are not dramatic (<1 km) but still significant in the ash plume region.

In summary, these findings indicate that using ICON instead of IFS as skin temperature source for VADUGS could slightly affect the results. Thus, even if it is preferred to use the same NWP model for the 24/7 operation as used for the training of the NN, the usage of ICON skin temperatures does not seem to modify the shape and VAMC or VATH values in a way that can significantly modify its meaning for aviation related services.

## 6 Conclusions

In the aftermath of the Eyjafjallajökull eruption in 2010 the VADUGS algorithm for the remote sensing of volcanic ash from thermal MSG/SEVIRI observations (Kox et al., 2013) has been developed at the Deutsches Zentrum für Luft- und Raumfahrt (DLR, German Aerospace Centre) using a machine learning approach leaning on a corresponding algorithm for ice cloud

remote sensing (Kox et al., 2014). This paper illustrates the VADUGS development and assesses the performance of the current version that is run operationally at the Deutscher Wetterdienst (DWD, German Weather Service) since 2015.

Unlike other spaceborne ash retrievals, VADUGS is based on a NN trained with simulated thermal satellite observations. In this approach the true ash properties (ash mass column concentration and ash cloud top height) to be retrieved by the algorithm are known exactly and the geographic and temporal distribution of the ash observations can be selected arbitrarily to allow for an application of the method to the entire MSG disk at each time of the day and of the night, and during each month of the year. For the creation of the training data set, the tool RTSIM has been implemented to automatically combine different

surface and atmospheric quantities based on historical numerical weather model results and other data sets as inputs for the radiative transfer simulation code libRadtran. The latter then produces realistic thermal observations for various meteorological conditions and with or without liquid and ice water clouds. RTSIM can be easily used and adapted to produce in future new additional training data sets.

       Ash detection is performed by setting a threshold to the retrieved mass column concentration: Pixels with BTD(10.8-12.0) $\leq$
$-0.6\,\mathrm{K}$ (pre-filtering) and retrieved VAMC $> 0.1\,\mathrm{g\,m^{-2}}$ are assumed to be ash contaminated. This results in a POD of 0.84 and a FAR of 0.05. Furthermore, VADUGS can detect 60 to 70 % of all ash loaded samples with true VAMC smaller than $0.1\,\mathrm{g\,m^{-2}}$. However, VAMC from VADUGS scatters considerably and errors (MAPE) between approximately 80 and 300 % are observed. Furthermore, the best results are obtained for moderate viewing zenith angles, corresponding to volcanic ash clouds in mid-latitudes. For VATH, a large scatter (3–5 km) is produced as well, although MAPE is always smaller than 70% and MPE is

even smaller (-10 to -20 % for viewing zenith angles smaller than approximately $50°$). Usually, low ash clouds with true VATH < 8 km are retrieved with higher correlations and smaller underestimations of -9 % for viewing zenith angles approximately between $40°$ and $50°$. However, when VAMC is large ($>2.0\,\mathrm{g\,m^{-2}}$), correlation usually increases for all clouds, thus enabling a better distinction between low and high ash clouds. This emerges also from the comparison with the spaceborne CALIOP lidar evaluated for the Puyehue-Cordón Caulle eruption in 2011. Here a reasonable correlation (0.49), MAPE (90 %), MPE (+55 %)

and RMSE ($0.41\,\mathrm{g\,m^{-2}}$) show that VADUGS is able to distinguish between thinner and thicker ash pixels although cloud top height is usually strongly underestimated. These results highlight the fact that the outcome of VADUGS depends on various factors and its accuracy can vary considerably. This can be observed also in the comparison to the standard 3-bands retrieval in Sect. 4.4.5, where both retrievals show different detection capabilities that are difficult to explain and have their origin in the complexity of the scenes observed. Furthermore, an additional comparison to airborne measurements by the FAAM (Facility

for Airborne Atmospheric Measurements, United Kingdom) and DLR aircraft in the context of the evaluation of the improved version of VADUGS (see below) shows that in that cases on those particular days VAMC is underestimated by VADUGS (see Piontek et al., 2021a, , their Fig. 13). Other retrievals, like the one in Prata and Prata (2012), can show different performances on similar data (overestimation, see Tab. 1 in their paper, with respect to FAAM measurements on the same day but at later times as in Piontek et al. (2021a)). These findigs point at the fact that remote sensing of volcanic ash is a serious challenge.

Every retrieval has its own strengths and deficiencies and provides an additional piece of information on the way to a more reliable and efficient mitigation of aviation hazards.

In general, the application of an ice cloud retrieval like in e.g. Kox et al. (2014) or Strandgren et al. (2017) is recommended in order to identify pixels where volcanic ash cannot be observed due to the presence of high clouds.

The principles of VADUGS can be applied for the creation of retrievals for other sensors with corresponding thermal channels, e.g. de Laat et al. (2020) adapted VADUGS to Himawari. Similar channels can also be found on the imager aboard GOES-R (Schmit et al., 2005), Fengyun-4A (Yang et al., 2016) or the upcoming Meteosat Third Generation spacecraft (e.g. Durand et al., 2015).

VADUGS is developed specifically for the Eyjafjallajökull 2010 eruption. This motivates the use of corresponding volcanic ash properties; specifically, only the refractive index of Eyjafjallajökull ash is applied. However, satellite retrievals are sensitive to the refractive index (Wen and Rose, 1994; Western et al., 2015; Ishimoto et al., 2021), which can vary significantly between different volcanos (Reed et al., 2018; Deguine et al., 2020). Just recently, methods to derive volcanic ash refractive indices of different ash types have been presented (Prata et al., 2019; Piontek et al., 2021c). Furthermore, ash microphysics (shape and size) also affects scattered and emitted radiation and can vary from one volcanic eruption to the other and also as a function of the distance from the source. Using the data set by Piontek et al. (2021c) constitutes a major improvement of the successor to VADUGS (Piontek et al., 2021b,a). The resulting retrieval also alleviates or eliminates other shortcomings of VADUGS mentioned in the course of the paper, thus encompassing for instance a larger temporal variability of atmospheric profiles and using more realistic radiative transfer simulations.

The operational application of VADUGS at DWD can thus provide airlines and other users with a useful spaceborne volcanic ash product that can be used to monitor volcanic ash evolution.

**Appendix A: Atmospheric profiles of clouds**

In IFS, a cloud-overlap algorithm computes the relative position of clouds across model levels and considers both cloud cover and cloud water content to realistically represent clouds on the spatial scale of the model (ca. 32 km for a spatial resolution of $0.25°$ in latitude and longitude). The maximum-random overlap assumption (Morcrette and Fouquart, 1986) used by ECMWF, particularly relevant for shortwave but also for longwave radiation, states that adjacent layers overlap maximally, while cloud layers separated by cloud-free layers overlap randomly. This procedure has to be translated into a 1D cloud structure (variability only in the vertical direction) since the 1D radiative transfer model can only handle cloud layers that are completely cloudy and because the spatial resolution of the satellite pixels (3 km at nadir, see Sect. 2.1) is a factor of 10 smaller than the model resolution. To this end, the layer-dependent cloud fraction is used to set up a random cloud structure consisting of 0 or more cloud layers, so that for each atmosphere layer a cloud fraction value of 0 or 1 is given. Adjacent cloudy layers are grouped and it is assumed that the cloud fractions overlap as much as possible in each group. The status of each layer is determined by the random position for an intersection through the corresponding group: All layers of the group are completely cloudy, if their cloud cover fraction exceeds the intersection position. Figure A1 shows an example for the determination of cloud layers. Two adjacent layers with cloud fractions of 37.5% and 50% make up the first cloud group. If the value of the intersection chosen at random is 25%, marked with a bold vertical line, then both layers have a higher cloud cover value and therefore are

625 cloudy, indicated by the bold layer number to the left. For the second group, with layers 5 to 8, three layers are cloudy for an intersection value of 50%, while layer 10, being the only one in the third group, has a value of 18.75% below an intersection value of 75%.

Cloud effective radii for each cloud layer are computed using two different parameterizations, one for water clouds and one for ice clouds. The applied water cloud parameterization is described by Bugliaro et al. (2011):

$$630 \quad r_{\mathrm{eff,L},i} = \sqrt[3]{\frac{0.75\,c_{\mathrm{L},i}}{Nk\pi\rho_{\mathrm{H_2O},i}}} \tag{A1}$$

$$k = \begin{cases} 0,8 & \text{for water} \\ 0.67 & \text{for land.} \end{cases} \tag{A2}$$

For layer $i$, $r_{\mathrm{eff,L},i}$ is the effective radius, $c_{\mathrm{L},i}$ the liquid water content, $N = 150 \cdot 10^6$ the droplet density, and $\rho_{\mathrm{H_2O},i} = 1000\,\mathrm{kg\,m^{-3}}$ the water density at 4°C.

Effective radii for ice cloud layers are determined by the parameterization from Wyser (1998); McFarquhar et al. (2003):

$$635 \quad \Delta T = \begin{cases} 273\,\mathrm{K} - T_i & \text{for } T_i < 273\,\mathrm{K} \\ 0\,\mathrm{K} & \text{else} \end{cases} \tag{A3}$$

$$b = -2 + 10^{-3} \left(\Delta T\,\mathrm{K^{-1}}\right)^{\frac{3}{2}} \log(20\,c_{\mathrm{I},i}\,\mathrm{kg^{-1}\,m^3}) \tag{A4}$$

$$r_{\mathrm{eff,I},i} = \frac{4}{\sqrt{3}+4} \left(377.4 + 203,3\,b + 37.91\,b^2 + 2.3696\,b^3\right)\,\mu\mathrm{m}. \tag{A5}$$

$T_i$ is the temperature and $c_{\mathrm{I},i}$ the ice water content. Typographyical errors in the formulas of both publications were corrected.

To avoid the implementation of perfect clouds, noise is introduced by the multiplication of the effective radii with uniform 640 random values between 0.9 and 1.1.

To determine the optical properties of water clouds the parameterisation by Hu and Stamnes (1993) is applied, with effective radius limit values of 2.5 $\mu$m and 60 $\mu$m. Optical properties of ice clouds are calculated using the parameterization of Yang et al. (2000); Key et al. (2002), extended by B. Mayer to cover the infrared spectral range (see Emde et al., 2016), for five ice particle habits (solid column, hollow column, aggregate, rosette-6, plate) encompassing the range of 2.85 to 108.10 $\mu$m. Ice 645 crystal shapes are selected randomly.

## Appendix B: Refractive index and optical properties of volcanic ash

The Earth Observation and Data group at the University of Oxford, and the STFC Rutherford Appleton Laboratory (RAL) Molecular Spectroscopy Facility (MSF) developed a set of analytical and experimental techniques to derive the refractive index from transmission spectra of laboratory aerosols. The method has been developed over many years; it has been applied to mimic 650 polar stratospheric clouds (Bass, 2003), sea salt aerosols (Irshard et al., 2009), Saharan dust (Peters et al., 2007) and later by Reed et al. (2017, 2018) to volcanic ash. The Aerosol Refractive Index Archive containing these spectral refractive indices

and other literature values can be found at http://eodg.atm.ox.ac.uk/ARIA/index.html. A short description of this method used to generate the Eyjafjallajökull refractive index used in this paper follows, more detail are contained in the references. The fresh Eyjafjallajökull sample used was collected at the ground by Dr. Evgenia Ilysinkaya in Iceland on 17 April 2010 at 18:20 local time downwind and ~ 6 km from the source. The sample is very fine as the eruption was phreatomagmatic at the time, and should be a good analogue to the long range transported ash at around this time (see also details of the aerosol cell given below).

A brief overview of the main experimental components is given in Fig. B1. The ash sample is first dried to remove any excess water and transferred for dispersal in a flow of nitrogen buffer gas. The aerosol is then injected into the 75 l aerosol cell and multi-pass optics are used to measure the optical transmission in the range 526–30,000 cm$^{-1}$ with a Bruker FTS and detector with a 1.9 cm$^{-1}$ resolution. A description of the aerosol cell can be found in McPheat et al. (2001), while the dispersal method that lofted the aerosol in the cell (particularly efficient for aerosol sizes smaller than 1 $\mu$m) is described in Reed et al. (2017). Aerosol size distributions are measured via different methods, an optical particle counter (radii 0.15–10 $\mu$m) and a scanning mobility particle sizer (radii 0.005–0.44 $\mu$m). For a detailed experimental description of the experiment, see Reed et al. (2017).

Gas absorption features in the infrared spectra are minimised by using a nitrogen buffer gas but some residual carbon dioxide and water lines still remain. These lines are removed by a line-by-line gas retrieval of the transmission spectra using the Reference Forward Model (RFM Dudhia, 2017), leaving the aerosol transmission signal.

In this experiment the forward model represents the aerosol cell transmission $T(\lambda)$:

$$T(\lambda) = \exp(-\beta(\lambda)x) , \tag{B1}$$

where $\beta(\lambda)$ is the volume extinction coefficient, $x$ the path length through the test cell at the wavelength $\lambda$. The extinction cross-section $\sigma_{ext}$ can be calculated, if we assume a particle scattering model and know the particle size distribution, from

$$\beta(\lambda) = \int\limits_0^\infty \sigma_{ext}(r, m(\lambda), \lambda) n(r) dr , \tag{B2}$$

where $r$ is the particle radius, $m(\lambda)$ the complex refractive index and $n(r)dr$ the number of particles per unit volume between radii $r$ and $r + dr$. A damped harmonic oscillator model is used to represent the complex refractive index $m(\lambda)$. Information from the sizing instruments constrains the size distribution used in the model. The complex refractive index in Fig. B2, where the real and imaginary part, $m_r$ and $m_i$ respectively, of the refractive indices are shown on top of the spectral response functions of MSG2/SEVIRI2, was retrieved from this model using the Levenberg-Marquardt method. Full details of the analysis method can be found in Thomas et al. (2005).

The refractive indices of volcanic ash presented here are a preliminary version of the data used in Ball et al. (2015); Reed et al. (2017, 2018) for Eyjafjallajökull ash. They assume Mie theory. Latter results in the Rayleigh regime use the continuous distribution of ellipsoids (CDEs) to account for non-spherical effects. These provided a better fit to the measurements but were not available at the time of this work. It should be noted that while CDE may provide better results, there are differences between the refractive index values in current publications which have arisen from differences in ash sample, analysis technique

and noise. In fact, $m_r$ from Reed et al. (2018) reaches below 1 like in the data set used here, but the minimum $m_r$ equals 1 for the measurement in Deguine et al. (2020). The maximum of $m_r$ remains smaller than 2 for Reed et al. (2018) but is as high as approximately 2.4 in Deguine et al. (2020), similarly to those used here. With respect to $m_i$, the peak in Fig. B2 is lower than in the data given by Reed et al. (2018) and Deguine et al. (2020).

Considering the spectral response functions of MSG2/SEVIRI2, Fig. B2 shows the expected "reverse" absorption feature (Wen and Rose, 1994; Pavolonis et al., 2006) mentioned in the introduction: Higher values of $m_i$ in the 10.8 $\mu$m channel with respect to the 12.0 $\mu$m channel indicate stronger absorption and thus smaller BTs in the first spectral interval than in the second one. Noteworthy is also the fact that absorption is higher at 8.7 $\mu$m than at 12.0 $\mu$m and that the maximum absorption takes place very close to the peak of the ozone channel centred at 9.7 $\mu$m.

Optical properties are computed with an early version of MOPSMAP (Gasteiger and Wiegner, 2018) based on the T-matrix code of Mishchenko and Travis (1998) and the improved geometric optics method (IGOM) code by Yang et al. (2007). Since volcanic ash is usually aspherical (e.g. Riley et al., 2003), we assume prolate spheroids with a distribution of aspect ratios from 1.2 to 5 with a median value around 2.1. Furthermore, two logarithmic normal size distributions are assumed with a standard deviation of 2. The distributions extend from 0.08 $\mu$m to 12.1 $\mu$m and their mode radii equal 0.4 $\mu$m and 2 $\mu$m, respectively. The size distribution range is thus in good agreement with literature (e.g. Chuan et al., 1981; Hobbs et al., 1981; Prata, 1989a; Turnbull et al., 2012). Mass density amounts to 2600 kg m$^{-3}$ (e.g. Wilson et al., 2012). Since the IGOM code doesn't support refractive indices smaller than one, the real part $m_r < 1$ of the refractive index had to be limited in the spectral range between approximately 8 and 10 $\mu$m to the supported range. In order to estimate the impact of this assumption on the computed extinction coefficients two calculations, one with cut and one without, have been performed for spherical particles. The result in Fig. B3 shows that only in the spectral range between 8 and 9.2 $\mu$m differences exist. Thus, only the 8.7 $\mu$m channel is affected and in this sensor band the inaccuracy of the mass extinction coefficient amounts to 1 % at most.

## Appendix C: Validation metrics

The probability of detection (POD) and the false alarm rate (FAR) are used as validation metrics to assess the accuracy of the detection performance of VADUGS, i.e. of VAC, while the mean percentage error and mean absolute percentage error are used to measure the accuracy of VATH and VAMC. These metrics are used later on in the validation section (Sec. 4.4).

The probability of detection (POD) measures how efficiently VADUGS detects volcanic ash. It is given by

$$POD = \frac{N_{TP}}{N_{TP} + N_{FN}} , \tag{C1}$$

where the number of true positives, $N_{TP}$, are all points correctly classified as ash, and the number of false negatives, $N_{FN}$, all missed ash clouds (see Tab. C1). The denominator, $N_{TP} + N_{FN}$, is thus the total number of points with ash clouds. The false alarm rate (FAR) measures how large fraction of the ash free points are falsely classified as being ash by VADUGS. It is given by

$$FAR = \frac{N_{FP}}{N_{FP} + N_{TN}} , \tag{C2}$$

**Table C1.** Contingency table for the ash detection from VADUGS.

|  |  | Truth | |
| --- | --- | --- | --- |
|  |  | Ash | No ash |
| VADUGS | Ash | $N_{TP}$ | $N_{FP}$ |
|  | No ash | $N_{FN}$ | $N_{TN}$ |

where the number of false positives, $N_{FP}$, are all points falsely classified as ash, i.e. the false alarms, and the number of true negatives, $N_{TN}$, all points correctly identified as ash free. The denominator, $N_{FP} + N_{TN}$, is thus the total number of points with no ash clouds. The corresponding simulation data (Sect. 3.1.7) are used as a reference when calculating the POD and FAR. Table C1 clarifies the quantities used to calculate the POD and FAR.

The mean percentage error (MPE) and mean absolute percentage error (MAPE) are used to measure the accuracy of the retrieved ash quantities retrieval with respect to the truth. The MPE is given by

$$\text{MPE} = \frac{100\%}{N} \sum_{i=1}^{N} \frac{E_i - T_i}{T_i}, \tag{C3}$$

where $T$ is the truth (Sect. 3.1.7, simulated observations), $E$ the estimated value by VADUGS and $N$ the number of simulations used. The MPE gives information about the direction of the deviations, i.e. whether VADUGS tends to overestimate or underestimate the values with respect to the simulated truth. The MAPE is given by

$$\text{MAPE} = \frac{100\%}{N} \sum_{i=1}^{N} \left| \frac{E_i - T_i}{T_i} \right|, \tag{C4}$$

and gives information about the average magnitude of the errors relative to the expected truth values. A value of 0.0 % means no deviation from the truth and a perfect correlation.

Finally, the standard deviation (RMSE, root mean squared error) of VADUGS with respect to the simulated truth is given by

$$\text{RMSE} = \sqrt{\frac{1}{N} \sum_{i=1}^{N} (E_i - T_i)^2}. \tag{C5}$$

*Data availability.* Please contact the authors for the data used in this paper or for additional data processed with VADUGS.

*Author contributions.* L. Bugliaro and D. Piontek have applied the retrieval, done the validation and written most of the text. S. Kox has implemented the first version of VADUGS as a NN. M. Schmidl has developed RTSIM. M. Vázquez-Navarro has contributed to the in-
stallation of VADUGS and the ice cloud retrieval at DWD and investigated possibilities to improve the retrieval. B. Mayer has developed

libRadtran and supervised its application for RTSIM. R. Müller has implemented VADUGS at DWD and performed the skin temperature sensitivity study. D.M. Peters and R.G. Grainger have measured refractive indices of ash. J. Kar has prepared the CALIOP data for validation. J. Gasteiger has computed optical properties of volcanic ash. All authors have contributed to the text.

*Competing interests.*  The authors declare no competing interests.

*Acknowledgements.*  We thank Hermann Mannstein (deceased) for his initial steps into machine learning in the early 2010s that led to various new satellite retrievals in the course of the years. We thank Kaspar Graf for his work with VADUGS and the establishment of VADUGS in the German community as a reliable volcanic ash algorithm. J. Kar thanks Mark Vaughan for useful discussions on correcting the low bias from the default ash lidar ratios used in CALIOP version 4.10. We thank two anonymous reviewers for a stimulating discussion leading to an improved version of the paper. L. Bugliaro, D. Piontek and M. Vázquez-Navarro were supported by the European Unions's Horizon 2020
research and innovation program under grant agreement no. 723986 (EUNADICS-AV). J. Gasteiger was funded by the European Research Council (ERC) under the European Union's Horizon 2020 research and innovation framework programme under grant agreement no. 640458 (A-LIFE).

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

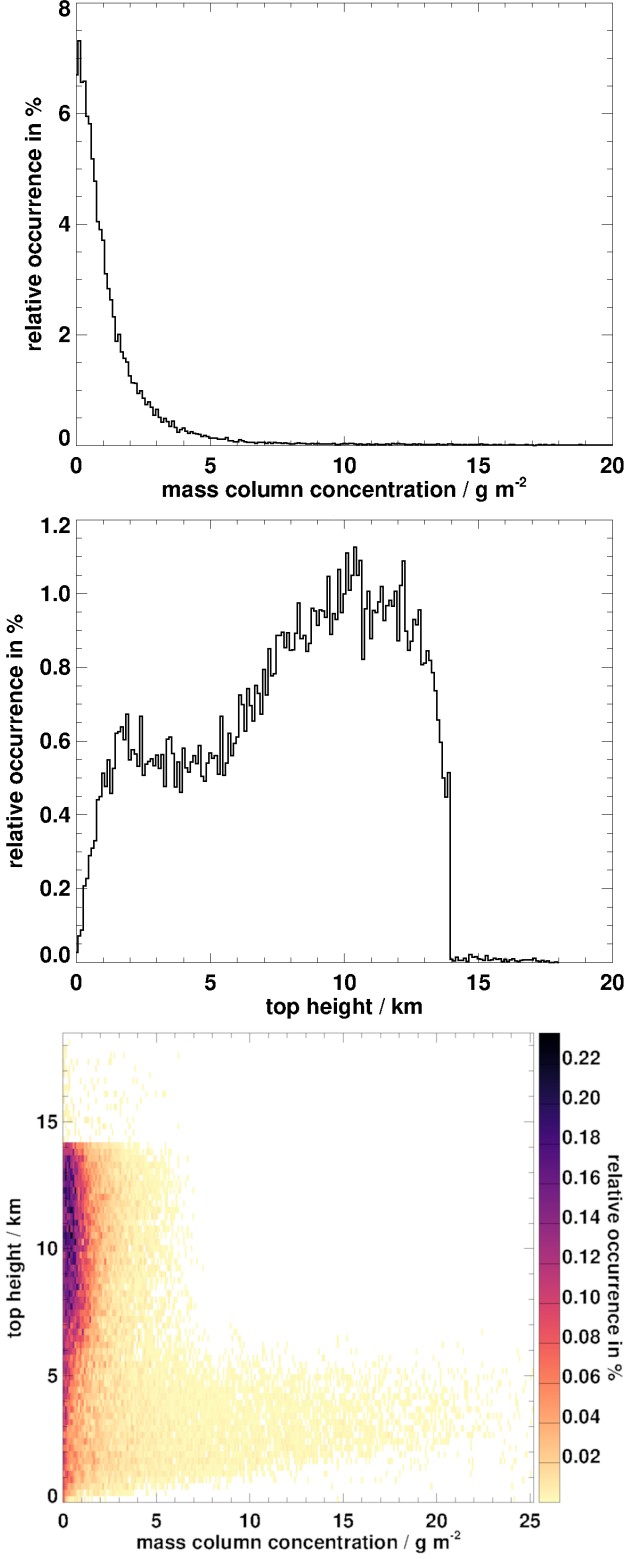

**Figure 1.** Histogram of ash concentrations and top height in the input data set used for the calculation of the training data set.

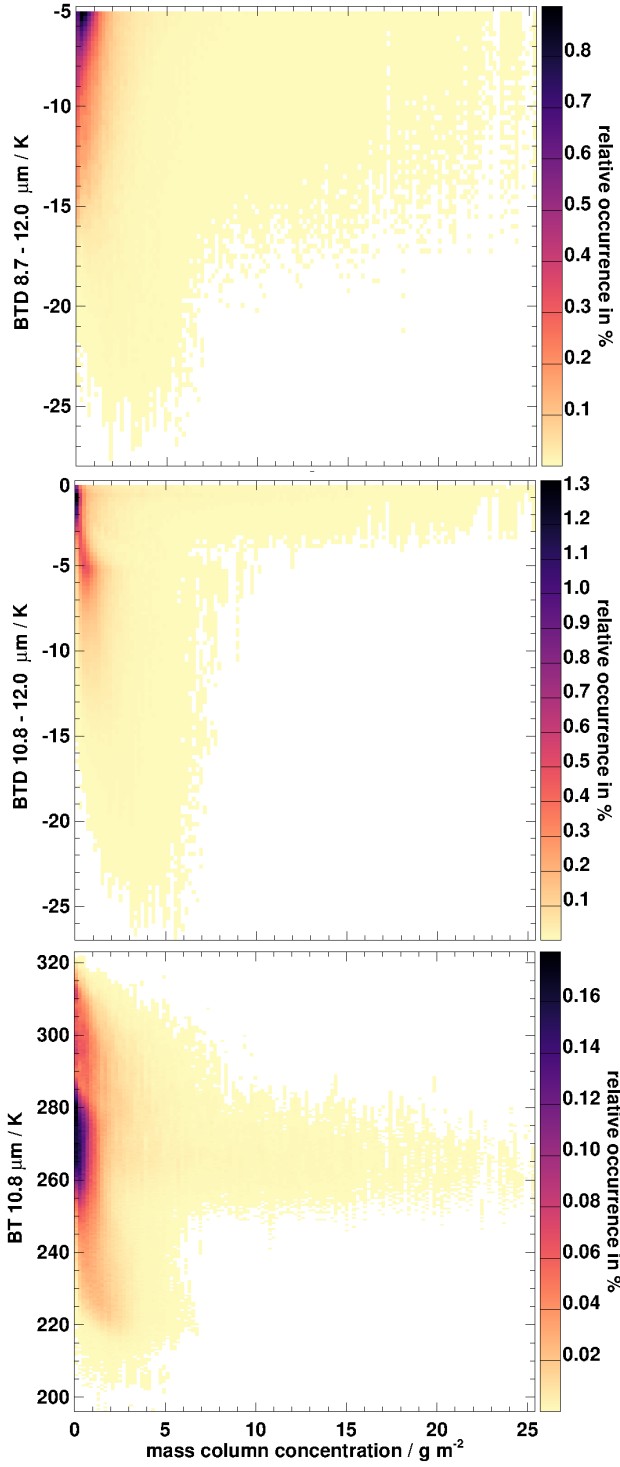

**Figure 2.** Histogram of (top) BTD(8-7-12.0), (middle) BTD(10.8-12.0) and (bottom) BT(10.8) for ash contaminated simulations as a function of VAMC.

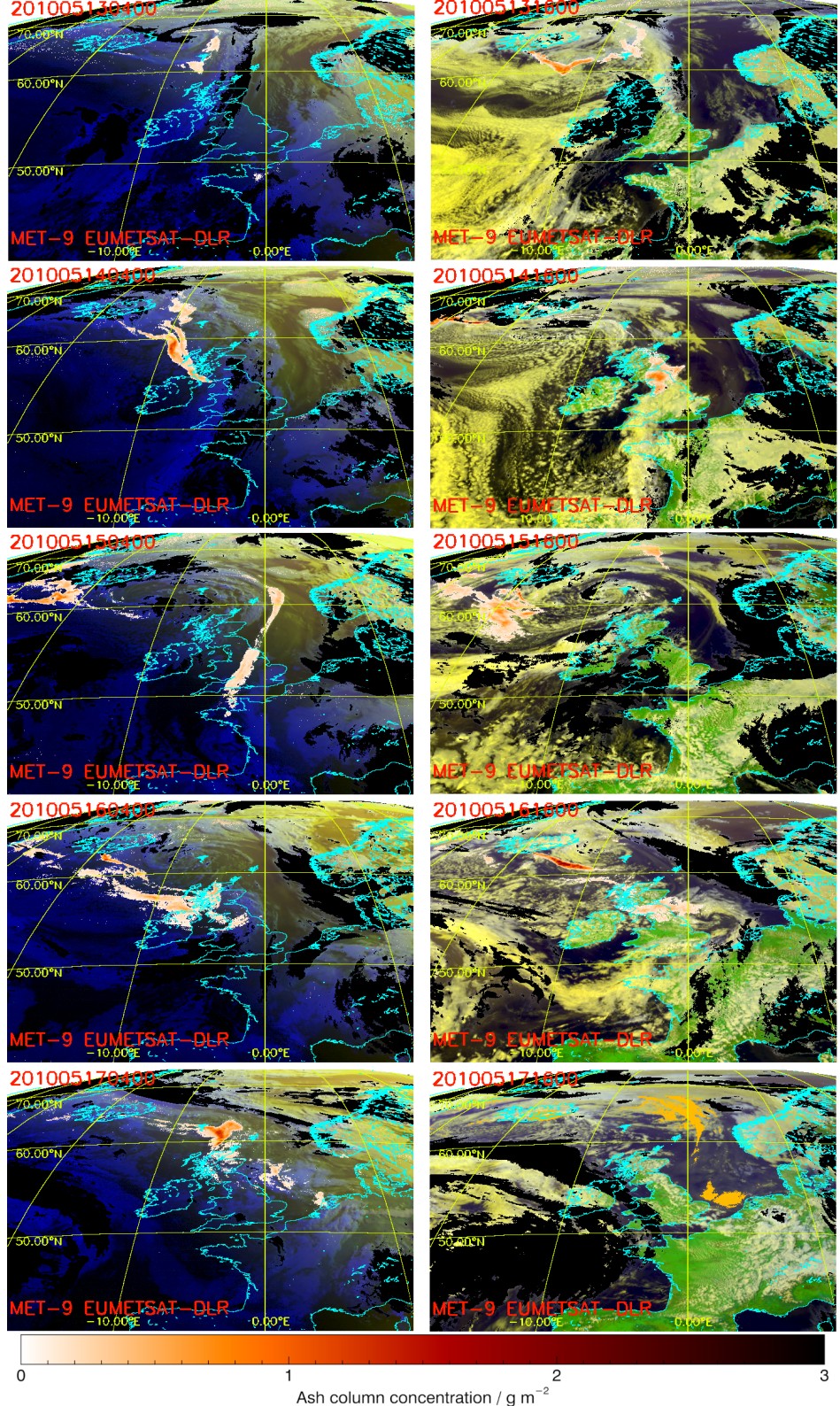

**Figure 3.** False colour RGB images with VAMC as white-red cloud on top (0–3 g m$^{-2}$) for the Eyjafjallajökull eruption in May 2010. Here, SEVIRI2 observations are shown between 13 May 2010 04 UTC and 17 May 2010 16 UTC in 12 h steps.

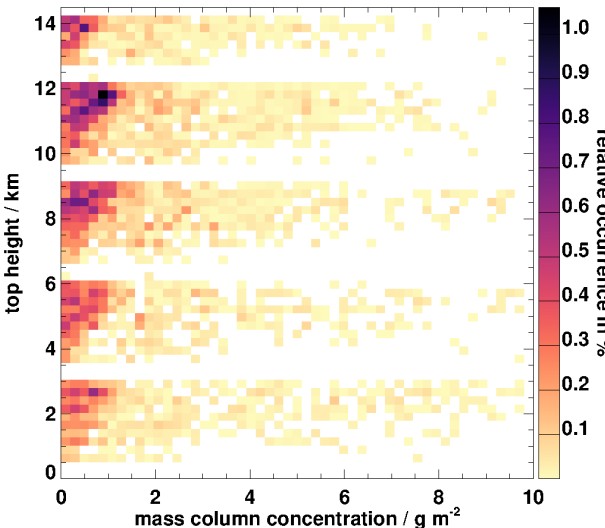

**Figure 4.** Two dimensional histogram of ash layer properties used as input for the calculation of the simulated validation data set.

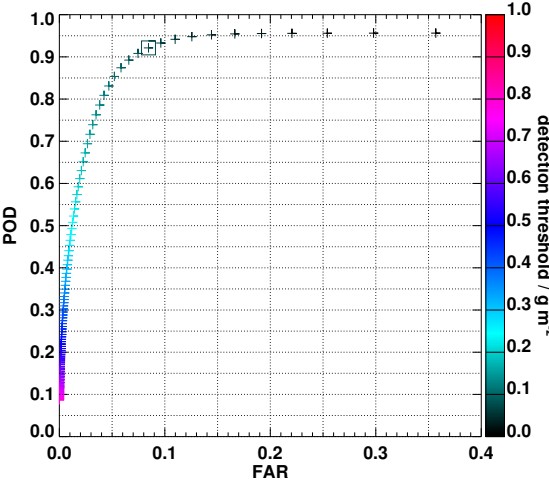

**Figure 5.** Probability of detection (POD) against false alarm rate (FAR) depending on the applied mass load threshold. The square marks the position corresponding to the threshold $0.1\,\mathrm{g\,m^{-2}}$.

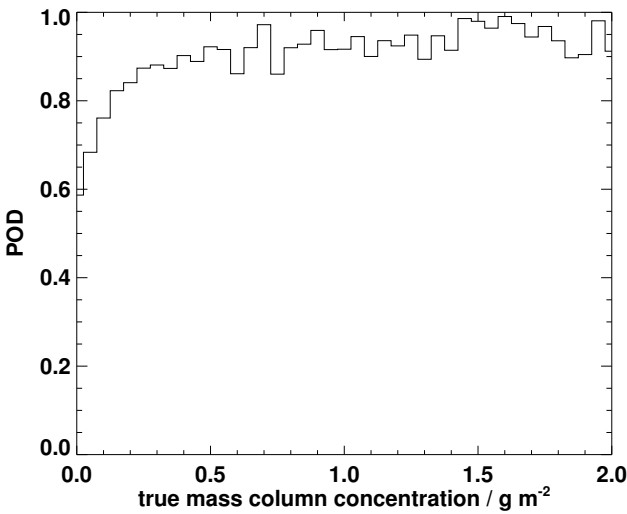

**Figure 6.** Probability of detection (POD) as a function of mass load after the application of Eq. 4 and of a threshold of $0.1\,\mathrm{g\,m^{-2}}$ to the retrieved VAMC.

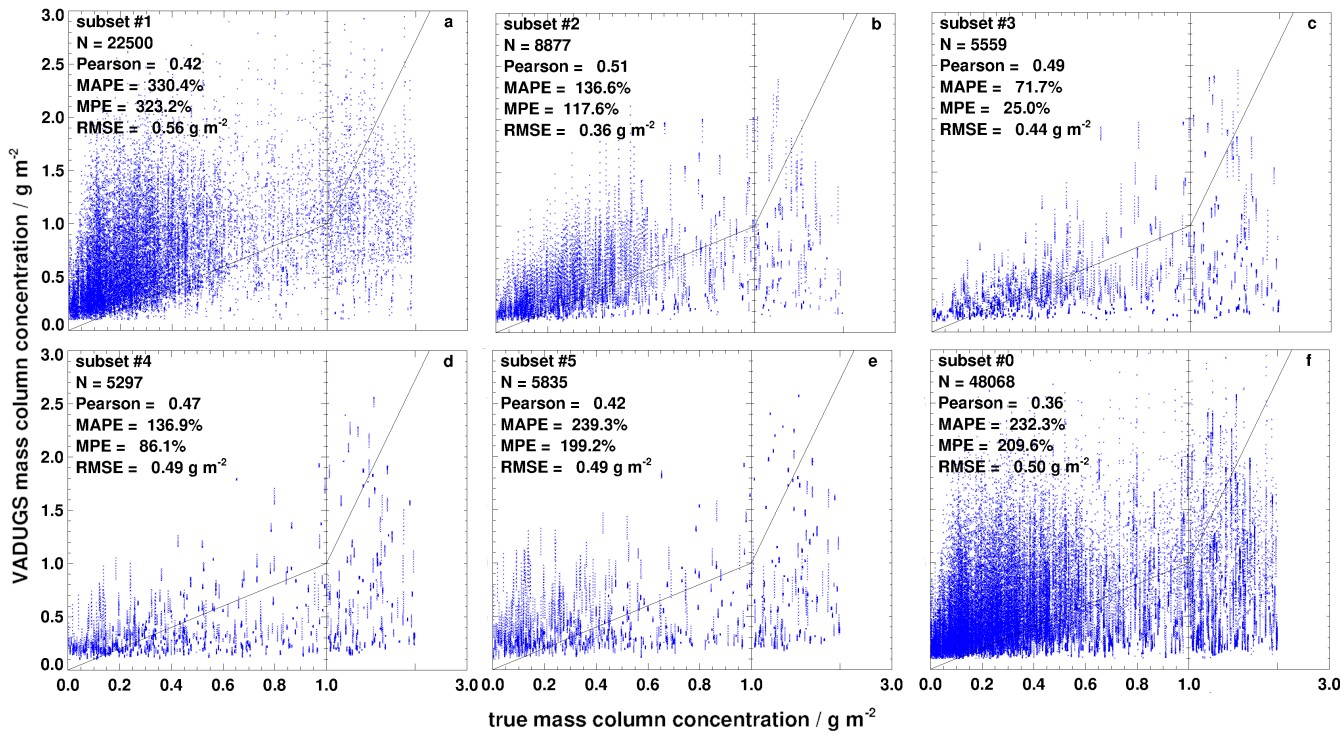

**Figure 7.** True against retrieved mass column concentration for the reduced simulated validation data set as a function of satellite viewing angle (subsets #1–#5). The solid black line shows the identity; for each data set the corresponding number of samples (N), the Pearson correlation coefficient, the root mean squared error (RMSE), the mean absolute percentage error (MAPE) and the mean percentage error (MPE) are given.

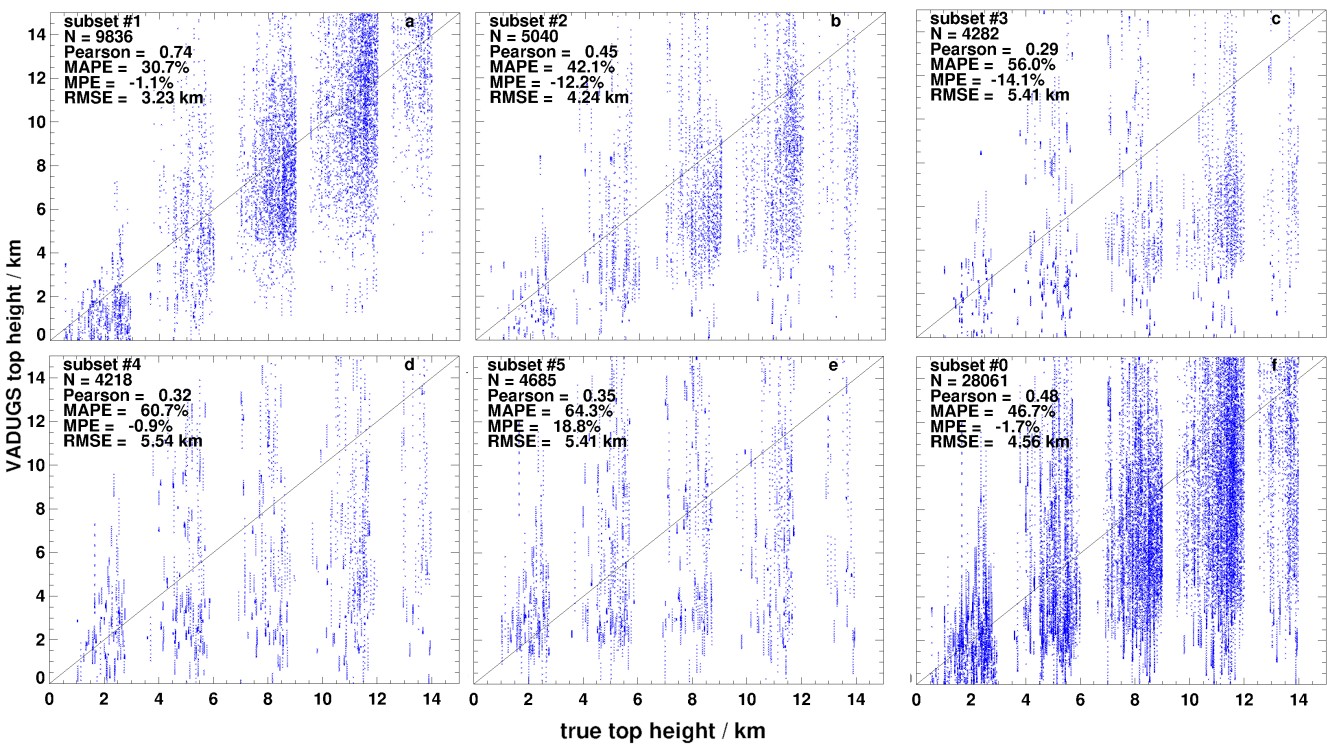

**Figure 8.** True against retrieved cloud top height for the reduced simulated validation data set as a function of satellite viewing angle (subsets #1–#5). The solid black line shows the identity; for each data set the corresponding number of samples (N), the Pearson correlation coefficient, the root mean squared error (RMSE), the mean absolute percentage error (MAPE) and the mean percentage error (MPE) are given.

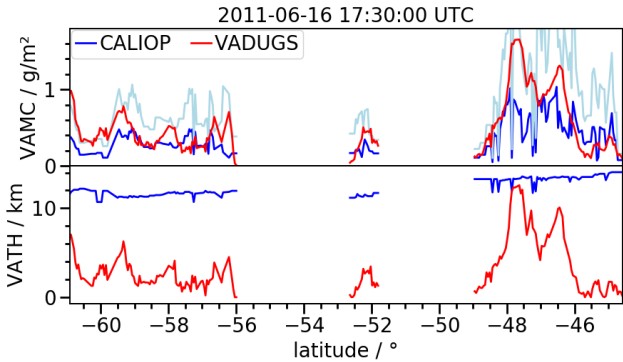

**Figure 9.** VAMC and VATH of a Puyehue-Cordón Caulle volcanic ash cloud around 16 June 2011 17:30 UTC as retrieved by CALIOP (blue) and VADUGS (red); the upper uncertainty of the CALIOP-retrieved VAMC is shown in light-blue.

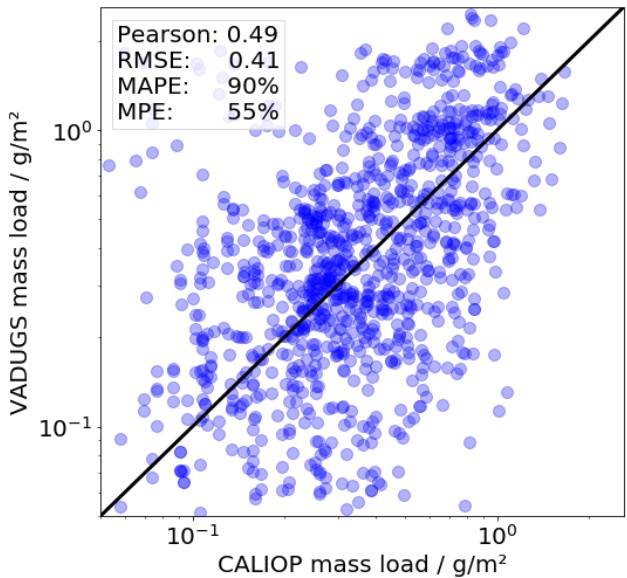

**Figure 10.** CALIOP and VADUGS retrievals of VAMC; the black line represents the identity.

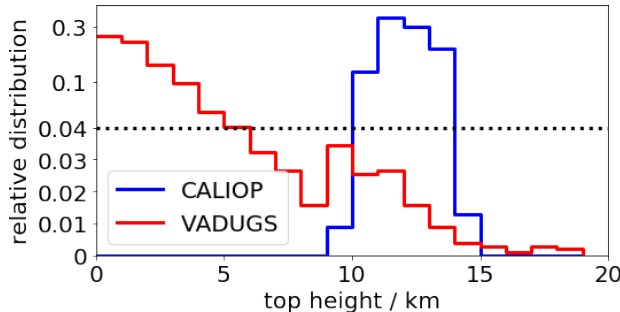

**Figure 11.** CALIOP and VADUGS retrievals of VATH; the vertical line separates the linear lower part from the logarithmic upper part.

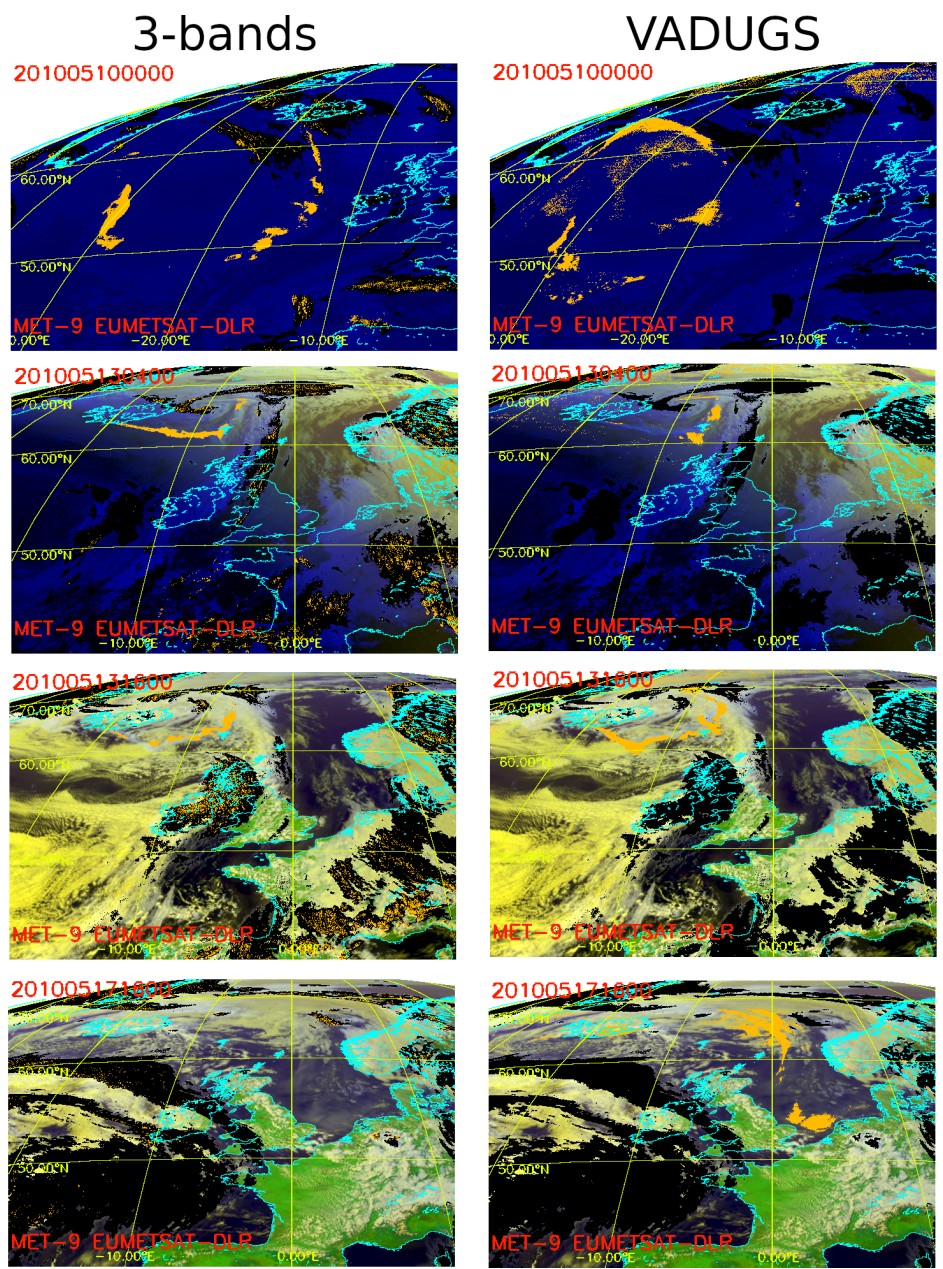

**Figure 12.** 3-bands and VADUGS retrievals of volcanic ash cover for four scenes of the Eyjafjallajökull eruption. The left column shows 3-bands volcanic ash masks in orange on top of a false colour RGB image, the right one VADUGS results. Black areas indicate the presence of ice clouds as obtained from COCS (see above), but ash detections inside ice clouds are not masked out as in Fig. 3 but shown on top.

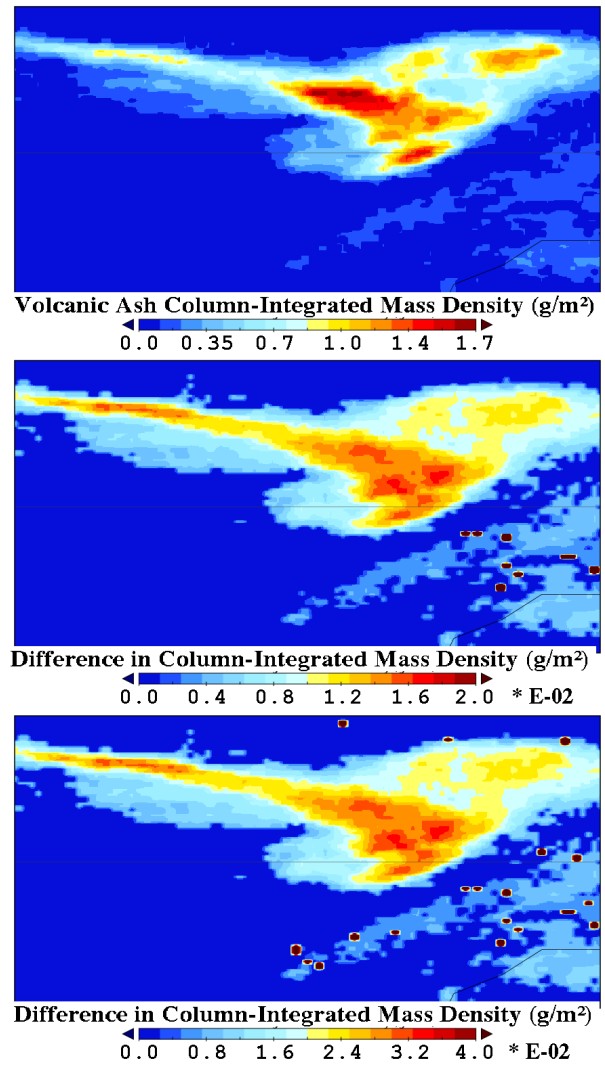

**Figure 13.** Top: VAMC output over the ocean South-East of Iceland in the triangle enclosed by the Faroe Islands in the South, the Shetland Islands in the West and the northernmost tip of Great Britain for 17 May 2010 01 UTC from the operational DWD processing chain. Centre: Difference in VAMC between the normal run and the run with surface temperatures increased by +1 K. Bottom: Same as in the centre panel, but with a skin temperature increase by +2 K.

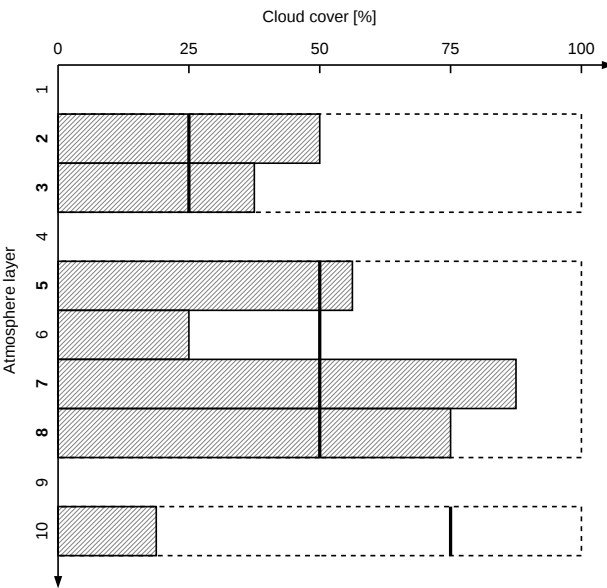

**Figure A1.** Scheme illustrating determination of cloud layers. Shaded boxes with solid borders show cloud layers with cloud cover percentages given by their horizontal extent. Adjacent cloud layers are combined to groups outlined by broken borders. For each group a bold vertical line marks the intersection position used for layer determination. Atmosphere layer numbers in bold indicate intersected cloud layers, which generate a cloudy layer during the setup.

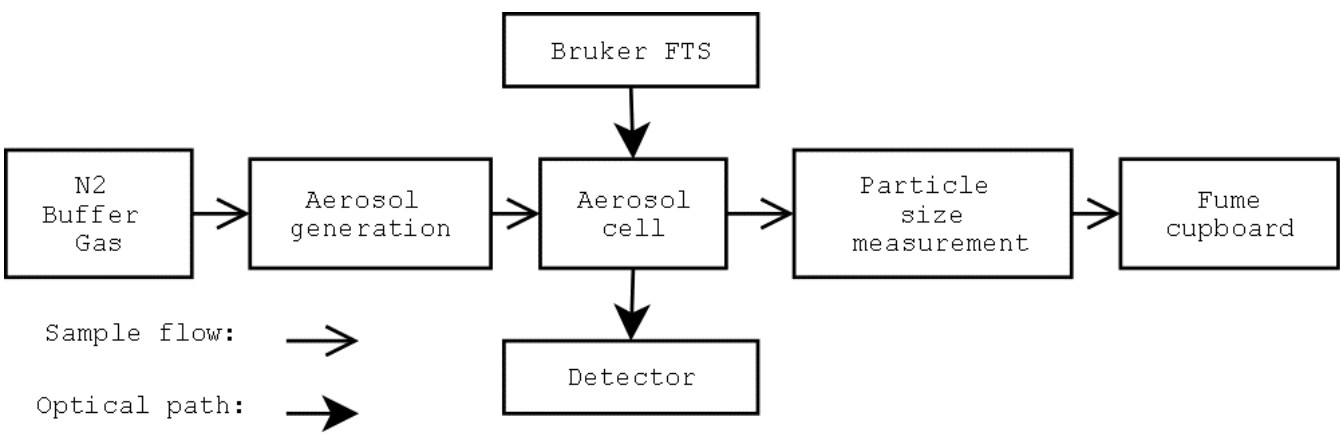

**Figure B1.** Simplified schematic of the experiment.

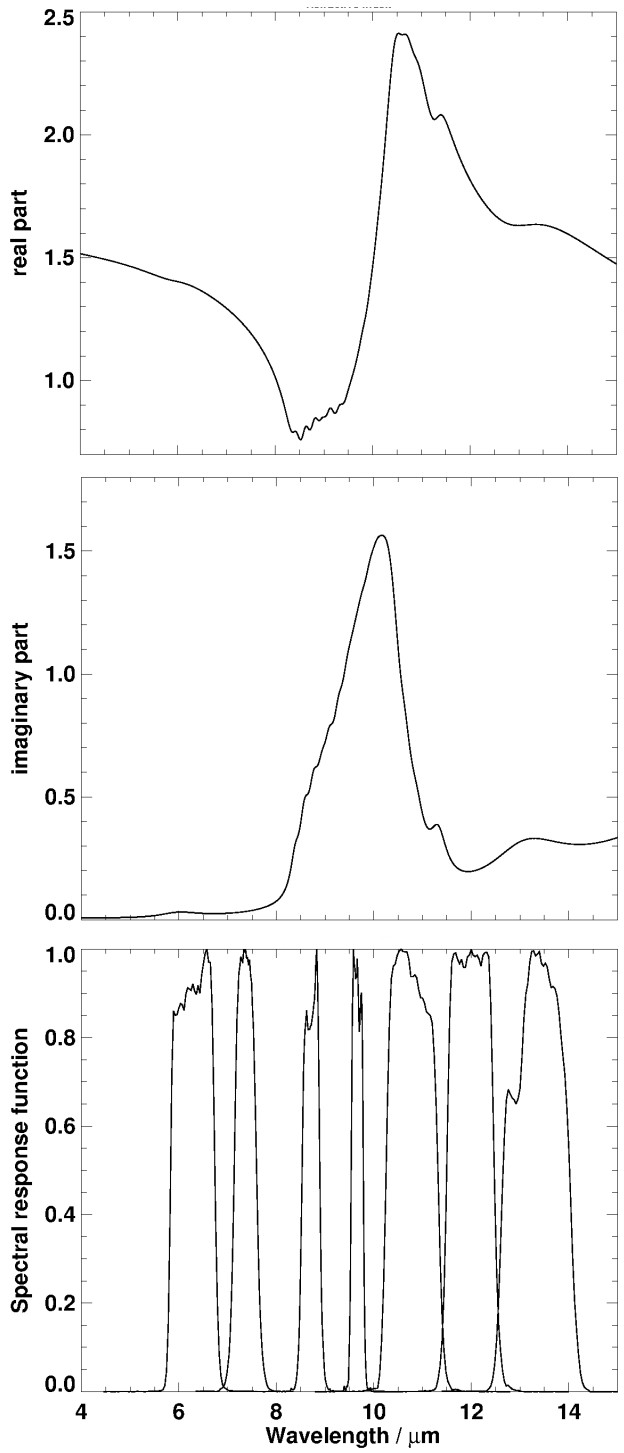

**Figure B2.** Real and imaginary part of the refractive index $m$ of Eyjafjallajökull ash (top two panels) and spectral response functions of MSG2/SEVIRI2.

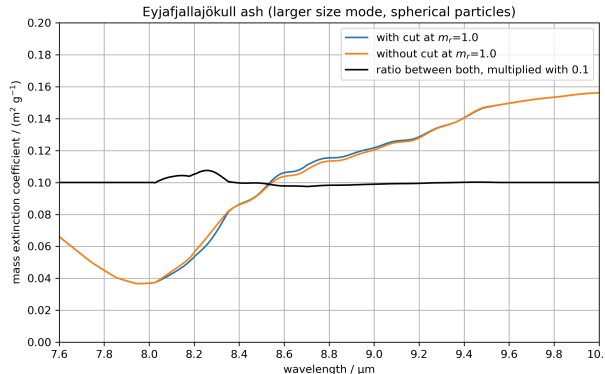

**Figure B3.** Mass extinction coefficients for spherical particles with the original refractive index $m$ from Fig. B2 (orange line) and with the same refractive index $m_r$ cut at 1 in the spectral region between 8 and 10 $\mu$m (blue line). The black line is the ratio between orange and blue line and has been multiplied by 0.1, meaning that a value of 0.1 corresponds to exact the same values in the two situations.