# Peer review of "VADUGS: A neural network for the remote sensing of volcanic ash with MSG/SEVIRI trained with synthetic thermal satellite observations simulated with a radiative transfer model"

_Natural Hazards and Earth System Sciences, 2021_

## Author Response (AR1)

**Answer to anonymous reviewer 1**

We thank reviewer #1 for his positive feedback! In the following we report his review comments in italic and provide answers to them directly after.

*Nonetheless, I feel like the whole manuscript could be simplified and shorten in order to better highlight the novelty of the presented results. The procedure used to simulate satellite observations and the VADGUS retrieval algorithm are well written but could be shorten significantly. I think that there are too many details that can be referred to the cited articles or by moving them to an appendix*

We have shifted Section "2.3 Validation metrics" to "Appendix C: Validation metrics", shortened "1.2 Atmospheric profiles of gases and clouds" considerably and moved a large part of these explanations to "Appendix A: Atmospheric profiles of clouds". Finally, we have transferred the text about the refractive indices used for the radiative transfer simulations to "Appendix B: Refractive index and optical properties of volcanic ash". The manuscript is now significantly shorter and it leads much faster to the validation part. For further changes please refer to the answer to reviewer #2.

**Answer to anonymous reviewer 2**

We thank reviewer #2 for his/her positive and detailed feedback! We have modified and improved the manuscript accordingly. Please notice in particular that the abstract and the conclusions have been adapted to take care of these comments. In the following we report the reviewer's comments in italic and provide answers to them directly after in plain text.

- *As stated by the previous reviewer, there are a number of sections, in particular relating to the production of the simulated observations where a lot of detail is provided. While this detail is critical for others to be able to reproduce the data, where possible detail that is also provided in references could be removed, and where this is not the case, the details could be moved to an appendix.*
  We have shifted Section "2.3 Validation metrics" to "Appendix C: Validation metrics", shortened "1.2 Atmospheric profiles of gases and clouds" considerably and moved a large part of these explanations to "Appendix A: Atmospheric profiles of clouds". Finally, we have transferred the text about the refractive indices used for the radiative transfer simulations to "Appendix B: Refractive index and optical properties of volcanic ash". The manuscript is now significantly shorter and it leads much faster to the validation part.
- *Title: Could 'radiative transfer calculations' be replaced by 'simulated radiances' or similar? To me there are multiple ways that radiative transfer calculations can and are used in volcanic ash retrieval and detection. The key use in this paper is to produce the simulated radiances for the training set. However, it is up to you which you think better summarises the more unique points in the paper.*
  We understand the point made by the reviewer and agree that a more specific title would help the reader catching the key issues of the paper already from the title. Thus, we have changed the title to "VADUGS: A neural network for the remote sensing of volcanic ash with MSG/SEVIRI trained with synthetic thermal satellite observations simulated with a radiative transfer model". It is a little bit longer but much clearer with respect to the content of the paper.

- *Abstract: Lines 8-14: Here the performance of the method of detection and retrieval is detailed. Can these values be compared to values using more conventional non neural network methods? There have been previous studies into this such as (Prata, A. J., and A. T. Prata (2012), Eyjafjallajökull volcanic ash concentrations determined using Spin Enhanced Visible and Infrared Imager measurements, J. Geophys. Res., 117, D00U23, doi:10.1029/2011JD016800.). These are primarily focused on the retrieval part rather than the detection. Most of these studies are with real data that have been validated against aircraft and/or Lidar data, which will be worth noting if comparing to validation completed against simulated imagery. For the detection part, could you provide some comparison with other leading volcanic ash detection algorithms, perhaps from one or both of the intercomparison workshops?*

  The reviewer's comment contains different aspects. On one side, he/she points out that mass column concentration from our retrieval has been mainly evaluated against simulated data sets. This is not completely true, since we presented in Section 4.4.4 a comparison to six overpasses of CALIOP, which is one of the instruments used in Prata and Prata 2012 as well as in other studies. This comparison is mentioned in the abstract, but we have now extended this part to make it clearer. This is a quite extensive amount of data such that we think that this already provides an exhaustive picture of the performance of VADUGS against independent data. Nevertheless, in the conclusions we have made a very qualitative comparison of the performance of VADUGS and the retrieval in Prata and Prata 2012 with respect to FAAM measurements (for VADUGS they are contained in the paper by Piontek et al. 2021c). On the other side, the reviewer suggests a comparison of the detection of ash with another non machine learning algorithm. We have added a section in the manuscript about this. We have selected the 3-bands method by Guéhenneux et al. 2015 that is used in real-time for the publicly available HOTVOLC monitoring system. It improves considerably the 2-bands method by Prata et al. 1989 and is easy to implement. Although VADUGS participated in both WMO volcanic ash comparison workshops, we don't have access to the data of other groups. The results are very interesting and improve the quality of the paper. The comparison shows that in many cases the ash clouds detected by the two algorithms are similar but different. Sometimes VADUGS detects ash close to the vent better than 3-bands, sometimes it is the opposite. In the examples shown, VADUGS seems to detect ash far away from the volcano better than 3-bands. These results point out that volcanic ash detection remains a challenge and that every remote sensing algorithm can provide additional information to improve the monitoring of ash.

- *Section 3.1.3, Lines 225-265: I feel this section is longer than necessary. It is not clear to me if the refractive indices used in this study were taken from work completed by others or if they are similar to those found by others but with small differences in method. If they are the same as those found in other published work, I think it would be better to reference the work with a statement saying followed the method of 'authors et al…'. Equally if the approach is similar to that of a previous study could the above approach be used but with the differences from that method stated. If the differences are significant could these be detailed in an Appendix.*

  The development of VADUGS has started quite early after the eruption of the Eyja volcano in 2010. At that time few data was available with respect to optical properties of ash in the thermal range. The results obtained from new measurements of the refractive index of Eyja ash were made available by the University of Oxford. In the course of the year, this data has been further evaluated and has been published in another form, such that the data used here cannot be referenced directly. Thus, we

have decided to keep the explanations but we have moved them to the appendix, as already mentioned above.

- *Lines 257-258: It seems a limitation of the training data to have used the refractive index from one eruption (at least in terms of the applicability to other eruptions). Given the dependency on refractive index and size distribution is strong it seems plausible that using values that are relatively well defined for Eyjafjallajökull may mean the scheme may be less accurate for other eruptions. I believe this limitation has been addressed in the successor version VACOS as mentioned in the conclusion. However, it would be good to state this here or perhaps when VACOS is first mentioned.*

  Yes, the usage of the refractive index from one eruption is a limitation, as already mentioned in the conclusions. We have added here two sentences to draw the attention of the reader to this point already here: "The refractive indices $m$ of volcanic ash are for Eyjafjallajökull as described in Appendix B. Please notice that this enables the retrieval to be tailored to this eruption, however the validation in Sect. 4.4.4 shows that VADUGS provides VAMC with a similar accuracy also for the Puyehue-Cordón Caulle eruption in 2011, thus indicating that its applicability could be extended to other volcanoes. Nevertheless, the usage of refractive indices for Eyjafjallajökull represents a principle limitation of VADUGS, that has been addressed by its successor (see Sect. 6)."

- *Line 294: "radiative transfer simulations are always run for a set of 41 cosines of the viewing zenith angles from 0.2 (corresponding to ~ 78 viewing zenith angle) to 1.0 (nadir)". I am not a neural network expert, but could this be part of the reason the method performs best at moderate zenith angle? One might expect it to perform best at low (or zero) zenith angle where viewing conditions are most optimum? I wonder if running simulations for 41 cosines of viewing angles might cause the neural network to favour scenarios that perhaps aren't likely, as presumably certain scenarios in terms of the atmospheric conditions will be more likely to occur at particular zenith angles given the fact that higher latitudes will all have relatively high zenith angle.*

  This is another interesting point. The simulation of many viewing zenith angles can be done with libRadtran in a very easy and fast way and increases the amount of simulated observations in a fast way and also increases the variability covered by our simulations. However, as you mention, some scenarios (meteorological conditions) are less likely than other for given viewing zenith angles. This might have a "confusing" impact on the NN that is difficult to quantify. In fact, it is difficult to predict for which viewing zenith angles the NN should work best. When looking at a very thin ash cloud the longer path through the atmosphere for large viewing zenith angles should be beneficial since it increases the effect of ash on brightness temperatures. For moderately thick ash clouds the situations might be different, since opaque ash clouds are expected to be more difficult to detect since they resemble water clouds more closely. We have added a sentence to the text: "This is meant to compensate for the fact that this approach might also lead to a more difficult learning procedure since not all meteorological conditions are observed for all viewing angles."

- *Lines 335-336: "To optimally cover different seasonal conditions we select day 15, 12UTC for 12 months from February 2010 to January 2011." What about impacts from different times of day, in particular inaccurate surface temperatures as the surface warms/cools? Again, this potential limitation may have been addressed in the successor and if so could be worth mentioning here.*

  These effects are only partly considered. 12 UTC is not local noon over the entire Meteosat disk such that with our approach that simulates many viewing zenith angles for every atmospheric column we have a mixture of observations at different local

times with the corresponding surface temperatures. However, nighttime simulations are not contained in the data set. In fact, this has been addressed by VACOS, and we have mentioned this now at these lines: "Although only one time of the day is used, different local times can be covered also due to the fact that many viewing zenith angles are simulated for every atmospheric column. Furthermore, VADUGS only relies on thermal observations such that the position of the Sun above the horizon is not relevant. Nevertheless, nighttime variability of e.g. surface properties like temperature cannot be accounted for (this has been improved for VADUGS successor)."

- *Lines 365-366: "Large VAMC > 5 gm-2 correspond to brightness temperatures between 260 and 280 K, thus corresponding to medium height levels up to approximately 5 km (see Fig. 4)." Does this mean there were no situations where VAMC > 5gm-2 occurred above 5km in the training set? This doesn't seem ideal given that high column loadings mostly occur close to the source when the plume is highest (often above 5km). If this is the case it would be good to explain in the paper why.*
  Our radiative transfer simulations have been organized in chunks and they have always been set up to reach as high as 14 or 18 km. At the same time the vertical extent was limited to 2.5 km and mass volume concentration was selected to be in different ranges, e.g. 0.001 to 10 mg m-3 thus spanning many orders of magnitude. After the simulations and the filtering procedure the resulting VAMC distribution is that described in Fig. 4. Although the data before the filtering covers all combinations of VAMC and VATH, the filtered data set is reduced in this respect such that VAMC larger than 6-7 g m-2 are not found at VATH larger than 7 km. This is of course a limitation especially close to the vent of the volcano, but as seen in the validation against CALIOP the values of VAMC found there are well below these values of 6-7 g m-2. Thus, we think that in most situations this does not represent a strong limitation for VADUGS. We have added a similar sentence to the text to address this aspect: "Although the data set before the filtering covers all combinations of VAMC and VATH, the filtered data set is reduced in this respect such that VAMC > 6-7 g m-2 are not found at VATH larger than 7 km at most. This represents a limitation especially close to the vent of the volcano, but as seen below in the validation against CALIOP (Sect. 4.4.4) the values of VAMC found there are well below these values of 6-7 g m- Thus, we think that in most situations this does not represent a strong limitation for VADUGS."

- *Lines 372-373: "Of course, other BTDs, like the most used BTD(10.8-12), can be implicitly obtained by the NN through combination of the available ones." Would it have made any difference to have used 10.8-12 or does the fact they are implicitly obtainable to the NN make it irrelevant which specific BTD's are used? In Kox et al 2014 it is suggested that the speed and accuracy of the neural network is dependent on the specific BT's/BTD's used. Could you explain why that may not be the case here or perhaps mention that some improvement could be found through use of different BTD's.*
  In fact, we have made some tests for CiPS (Strandgren et al. 2017a, our ice cloud retrieval) where we have seen that the BTDs as input eventually do not change the accuracy of the resulting NN. But since the physics is contained in the BTDs it is nice to have them as input. In addition, using input quantities that are directly related to the output quantities can affect the speed at which the training converges. Relationships can be obtained faster by the NN when the BTDs are used. But even this aspect was not really evident with the neural network framework we used for CiPS, which was in C, but it was more important back in 2012 when COCS Kox et al. 2014) was first developed with self-made IDL routines.

- *Lines 384-385: "Surface emissivity is neglected here since its variability is not very strong." Ashpole et al (2012) https://agupubs.onlinelibrary.wiley.com/doi/pdf/10.1029/2011JD016845 suggested that spatial variability in the surface emissivity can cause issues in the detection of dust (and by extension ash). Do you have a reference or could you explain why the variability is not considered sufficient to be significant here?*

  There is spatial and temporal variability in surface emissivity, you are right, and we consider it when we do the radiative transfer calculations. However, we decided not to use surface emissivity as input to the NN since on one hand the NN should have learned something about it during training and on the other hand because we think that the main impact on SEVIRI brightness temperatures comes from surface temperature, clouds, ash and water vapour. Furthermore, as already mentioned in the paper, it is difficult to obtain daily/hourly resolved spectral surface emissivity values, especially for near-real time applications. We have specified the text accordingly: "Surface emissivity is neglected here since its variability is thought to be of secondary importance compared to surface temperature and because it is difficult to obtain daily/hourly values of this quantity, especially for a possible near real-time application."

- *Lines 451-452: "With the help of this additional filtering, the overall POD sinks to 0.84 but the FAR also sinks to 0.05." Is this an improvement on 0.92 and 0.08? Could you explain why reducing FAR appears to have been prioritised over maintaining POD? As mentioned earlier, if this scheme was used in either of the intercomparisons it would be good to discuss its relative performance.*

  The false alarm rate (FAR) measures how large fraction of the ash free points are falsely classified as being as Since the vast majority of pixels is ash free (in contrast to ice/water clouds), we thought that a lower FAR is beneficial. Furthermore, one of the objections against the VAACs in 2010 was that they had closed a to large part of the air space. In order to avoid this, a lower FAR is beneficial as well (at the costs of course of POD). Finally, the first tests of VADUGS with real data after the training of the NN with simulated brightness temperatures gave us the impression that FAR would be a problem. But yes, this is the standard procedure we have always applied in all sections following this one. We have added a remark about this: "Even if the gain in FAR (FAR measures how large fraction of the ash free points are falsely classified as being ash, see Appendix C) is low, this can result in many pixels being misclassified as ash, which would artificially enlarge the area covered by ash that should be avoided by air traffic and is thus preferred to a larger POD. This filtering is always applied in the rest of the manuscript, and in all other applications as for instance at DWD (Sect. 5)."

- *Lines 553-554: "The distribution of VADUGS-retrieved VATH peaks at 0 km and has a flank reaching up to 19 km, with a notable minor peak at about 9 to 12 km." Do you have any thoughts why the main peak is around 0km? This kind of ash should be hard to detect. Is this a result of there being little training data very close to 0km or perhaps there being lots of training data between 0 and 5 km? Could you explain why the training data appears to sharply reduce in frequency above 14km? Large eruptions can emit ash higher than this, although if the focus of the scheme is on tropospheric ash then limiting the training data above the tropopause may be wise. Also it may be worth mentioning that the general underestimation of VATH is likely related to (or perhaps the cause of) the typical overestimations in VAMC.*

  The real VATH is represented by the blue curve, obtained from CALIOP. The red curve is VATH from VADUGS, which in many cases is not able correctly retrieving it. Thus, the problem here is not retrieving VATH for low ash clouds but for high ash

clouds, close or above the tropopause (CALIOP shows almost all clouds with VATH > 10 km and many clouds with VATH > 12 km up to 15 km). VADUGS always underestimate height, but under these circumstances it seems that in many cases it is not able to retrieve high VATH, probably because of the unusual combination of brightness temperatures induced by the ash in the tropopause region, where water vapour is very low, temperature is constant or increases again with height. This region, for these reasons, is not well covered in the training data set. As far as the distribution of VATH and VAMC in the training data is concerned, please refer to one of the previous questions/answers. We have added a sentence about the relationship between VATH and VAMC: "Notice that the general overestimation of VAMC is again related to (or maybe induced by) the underestimation of VATH."

- *Figure 7: Could the gaps at certain height levels (i.e. every 3km) be explained?*
  As for the training data set, we have simulated the data set for validation in various chunks, with each of them covering a different height range. These ranges are visible here. We have modified the corresponding lines to better explain this: "For each ash profile samples with and without volcanic ash/meteorological clouds are simulated in various height ranges (tops between 0.5 and 14 km), thus producing the gaps observed in Fig. 4. After application of the filter implemented in Sect. 4.1 with Eqs. 1, the data set comprises 3,526,397 samples with 100,083 ash loaded samples (those plotted in Fig. 4). Most points accumulate to concentrations below 1 g m-2 and only few samples at higher concentrations."

**Technical corrections**

*Line 131: "are all points correctly classified as ash and the number of false negatives, NFN, all missed ash clouds (see Tab. 1)." I think either a comma after the first use of 'ash' above would help or 'NFN, are all missed ash clouds.'*
Done, thank you very much, it was misleading without comma!

*Line 136-137: "are all points falsely classified as ash, i.e. the false alarms and the number of true positives, NTN, all points correctly identified as ash free." As above and also 'true positives' should be 'true negatives'.*
Corrected.

*Line 372: 'implicitely' should be 'implicitly'*
Corrected.

*Line 375: 'channels' should be 'channel'*
Corrected

*Line 380: 'induce' should be 'deduce'?*
Yes, thank you.

*Line 429: 'For each' I think a comma after each would improve readability*
It was thought to be "For each ash profile, samples…".
Corrected.

*Line 430: 'profiles' should be 'profile'*
Corrected.

*Line 543: 'bahaviour' should be 'behaviour'*
Corrected.

*Lines 621: 'been' should be 'be*
Corrected.